# A tube-source X-ray microtomography approach for quantitative 3D microscopy of optically challenging cell-cultured samples

Ilmari Tamminen[1✉], Kalle Lehto[2], Markus Hannula[1], Miina Ojansivu[3], Laura Johansson[2], Minna Kellomäki[1], Susanna Miettinen [1], Antti Aula[1,4], Teemu Ihalainen [1] & Jari Hyttinen [1]

Development and study of cell-cultured constructs, such as tissue-engineering scaffolds or organ-on-a-chip platforms require a comprehensive, representative view on the cells inside the used materials. However, common characteristics of biomedical materials, for example, in porous, fibrous, rough-surfaced, and composite materials, can severely disturb low-energy imaging. In order to image and quantify cell structures in optically challenging samples, we combined labeling, 3D X-ray imaging, and in silico processing into a methodological pipeline. Cell-structure images were acquired by a tube-source X-ray microtomography device and compared to optical references for assessing the visual and quantitative accuracy. The spatial coverage of the X-ray imaging was demonstrated by investigating stem-cell nuclei inside clinically relevant-sized tissue-engineering scaffolds (5x13 mm) that were difficult to examine with the optical methods. Our results highlight the potential of the readily available X-ray microtomography devices that can be used to thoroughly study relative large cell-cultured samples with microscopic 3D accuracy.

[1] BioMediTech, Faculty of Medicine and Health Technology, Tampere University, Tampere, Finland. [2] BioMediTech, Tampere University of Technology, Tampere, Finland. [3] BioMediTech, University of Tampere, Tampere, Finland. [4] Department of Medical Physics, Imaging Centre, Tampere University Hospital, Tampere, Finland. ✉email: ilmari.tamminen@tuni.fi

Cell-cultured constructs are widely developed and studied for a broad range of purposes. We believe the scientists would benefit from quantitative tools that can be used to assess the cell populations inside these complex 3D structures. For example, proper differentiation and proliferation of stem cells inside tissue-engineering scaffolds and the overall safety of these medical constructs need to be verified comprehensively. As another example, it is important to assess the cell responses in organ-on-a-chip platforms, used in drug-administration studies. Therefore, visual examination and quantification of features such as the distribution, interactions, morphology, antigen expression, and differentiation of cells in these constructs are essential. Furthermore, statistical analyses become more reliable if larger samples can be analyzed. Thus, cell-cultured constructs should be assessed as large 3D biological communities instead of inspecting individual cells or a few cells at a time.

Light microscopy has multiple good properties such as sub-micron resolution, low phototoxicity, and spectral differentiation of several signals simultaneously. However, the main drawback, even with advanced imaging techniques that rely on conventional optics, is that the depth range is currently limited to around a couple of millimeters in turbid samples even without strong absorptive properties (Supplementary Note 1). Depending on the sample type, the depth range of 3D microscopy can be increased by various sample preparation techniques. Solvents[1] and ionic detergents with electric fields[2] can be used to equalize the refractive-index mismatches and to remove scattering lipid structures, respectively. However, refractive-index matching is not applicable for biomedical materials that are inherently too photon deflective or absorptive, and with lipid extraction methods are best suited only for certain types of histological samples. Another way to tackle the photon deflection is to use adaptive optics that can be used with techniques such as multiphoton fluorescence imaging and optical coherence tomography[3]. Unfortunately, few scientific institutions currently have access to these usually custom-made imaging systems. Although microscopic focuses have been achieved in depths of millimeters in turbid samples[4], to the best of our knowledge, no optical 3D imaging modality has been demonstrated to be able to resolve cellular details at depths deeper than those achieved by conventional optics. In addition to these problems, low-energy imaging techniques might be limited to a particular sample and label combinations to minimize absorbance losses.

By increasing the imaging energy, it is possible to perform straightforward 3D microscopy with a good depth range that is tolerant of the varying optical features of different sample types. X-ray imaging provides an efficient way to acquire 3D images of practically any type of samples, including both optically deflecting and absorbing materials. X-ray imaging technology is also commercially well-established and hence available for a broad scientific community. Usually, the studied features are distinguished according to their X-ray attenuation characteristics. In addition to native contrast, a variety of chemical and immunological X-ray contrasting methods are available for image enhancement (Supplementary Note 2). One common category of the X-ray 3D microscopes is the tube-source X-ray microtomography (μCT) devices. Although X-ray tube source μCTs are readily available, relatively small, and inexpensive, their accuracy and image quality are considered to greatly limit the subsequent analyses when compared with more advanced X-ray technologies such as synchrotron-based imaging setups (Supplementary Note 3). However, novel and useful ways to apply all pre-existing technologies efficiently should be considered and experimented—only this way, the scientific community can efficiently utilize the available resources.

Instead of relying on the more advanced but less available X-ray technologies, we demonstrate the potential of a commercial, tube-source μCT device (Xradia MicroXCT 400, Carl Zeiss). We describe practical principles in detail on how to visualize and quantify cellular structures in 3D with good spatial coverage and large FOVs enabled by the micro-focus tube-source X-rays. Antibody-silver labeling is used to enhance the studied cell structures in otherwise low-density samples. The visual and quantitative accuracy of the μCT imaging is assessed with comparison to optical reference methods. The spatial coverage of the μCT imaging is demonstrated by investigating stem-cell nuclei inside clinically relevant-sized tissue-engineering scaffolds that are challenging to assess with the optical methods. We used the same samples to demonstrate how induced cytoskeletal relaxation of the nuclei can be quantified from the 3D μCT data. After demonstrating the spatial coverage and assessing the data flow of a large number of particles, we made a detailed quantitative 3D comparison of our μCT method to optical 3D reference data. The quantitative analyses are founded on a reference-based adaptive segmentation of the cell nuclei, data which is processed by a newly developed 3D-Round-Object-Quantification Algorithm (3DROQA, Supplementary Methods, Protocol 1). Our results highlight the potential of the readily available tube-source μCT devices that can be used to visually and quantitatively study cell structures in relatively large biomedical constructs thoroughly in 3D with microscopic accuracy.

## Results

**Visual comparison of cell structures on flat substrates.** First, we verified the ability of the used X-ray tube-source μCT to visualize a variety of antibody-silver labeled subcellular structures (Fig. 1, the principle of the labeling is illustrated in Supplementary Fig. 1, the entire μCT imaged flat-substrate surfaces are shown in Supplementary Fig. 2). Fluorescence, bright field, and confocal laser scanning microscopes (CLSM) were used to acquire optical reference data. Human adipose stem cells (hASC) and epithelial Caco-2 cells were seeded and labeled on well-plate-based flat substrates. Primary antibodies against lamin A/C, β-actin, ATP5α, and ZO-1 were used to label nuclear lamina, actin cytoskeleton, mitochondria, and cell-cell junctions, respectively. To follow the proper development, the silver generation of lamin A/C labeled hASC in a separate experiment was recorded with low-energy red-light microscopy (Supplementary Movie 1). No background staining was observed even after exceeding the maximum generation time recommended by the manufacturer of the silver-kit (#6010 EnzMet™ for General Research Applications, Nanoprobes). Taking into account the optical blur, the expected nuclear circumference of an example nucleus, faintly visible before the generation, matched with the distribution of the final silver signal, as demonstrated in the Supplementary Movie 1. The intensity varied between the cells, likely due to varying lamin A/C expression levels.

In the main experiments, the acquired silver signals were compared with similarly localizing fluorescent labels such as 4',6-diamidino-2-phenylindole (DAPI) for nuclei. The X-ray and optical images showed high morphological similarity (Fig. 1). In particular, the μCT-imaged lamin A/C, actin, and ZO-1-labeled cells all exhibited sharp and intensive details with easily recognizable morphologies. In the μCT data, weak metal streaking artifacts were seen to project from the strongest nuclear signals. These artifacts were possibly enhanced further by the straight substrate surface[5]. The spacious nuclei were also reproduced in the anti-β-actin μCT data, as roundish, low-intensity features where the labeled cytoskeletons do not extend. The grainy essence of the silver became apparent when the light microscopy images of the anti-β-actin-labeled cells

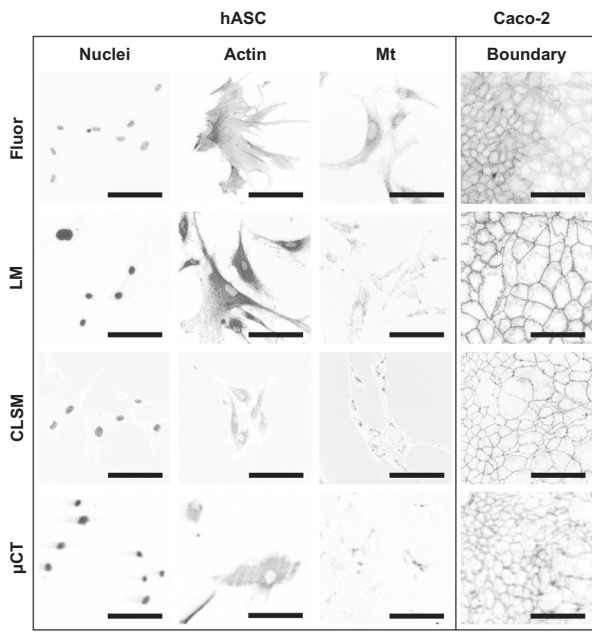

**Fig. 1 Visual comparison of cell structures on flat substrates.** Cells seeded on well-plate-based flat-substrate samples were used to assess the visual accuracy of the tube-source μCT imaging with comparison to optical references. The nuclei of hASCs were fluorescently labeled with DAPI (hASC-Nuclei-Fluor), actin cytoskeleton with phalloidin label (hASC-Actin-Fluor), and mitochondria with MitoTracker Red (hASC-Mt-Fluor). Caco-2 cells were also labeled with the phalloidin label to adduce the honeycomb-resembling cell boundaries exhibited by the orbicularly arranged actins (Caco-2-Boundary-Fluor). Due to the attenuating properties of the dark silver precipitates, separate hASC and Caco-2 cells were antibody-silver labeled by using antilamin A/C (hASC-Nuclei), β-actin (hASC-Actin), ATP5α (hASC-Mt), and ZO-1 (Caco-2-Boundary) primary antibodies (three lower image rows). All the antibody-silver labeled cells were observed with a light microscope (LM), CLSM in reflection detection mode (CLSM), and μCT (μCT). Darker tones represent higher signal intensities. The μCT voxel size was 1.2 μm. Full μCT-imaged surfaces without histogram adjustment and grayscale inversion are available in Supplementary Fig. 2. All scale bars are 100 μm.

were compared with the phalloidin signals having filamentous, more representative fine morphology. Taking into account the clear background, we believe the grainy texture is not a result of background staining; it could be the inherent property of the label itself. In any case, the grainy texture was much smaller than the μCT voxel size of 1.2 μm, and hence more accurate labeling would not provide any extra information in the μCT data. The fine curly ZO-1 features seen by the light microscope were not reproduced in the μCT data. Ideally, according to the Nyquist sampling theorem and the used voxel size, at least the largest curls should have been resolvable but were possibly smeared out by different image artifacts[5,6]. As expected, the ATP5α signals were also localized into irregular mitochondrial networks, but were, however, the most difficult features to distinguish from the μCT data. If needed, the finest features of mitochondria and ZO-1 are likely better resolved by using nano-CT devices[7], but compromises with other imaging properties such as decreasing the size of the FOV are likely necessary. Details about the Fig. 1 image composition are given in Supplementary Note 4.

**Visual comparison of nuclei in scaffolds**. We wanted to investigate the potential of the tube-source μCT in visualizing cellular features inside clinically relevant-sized tissue engineering

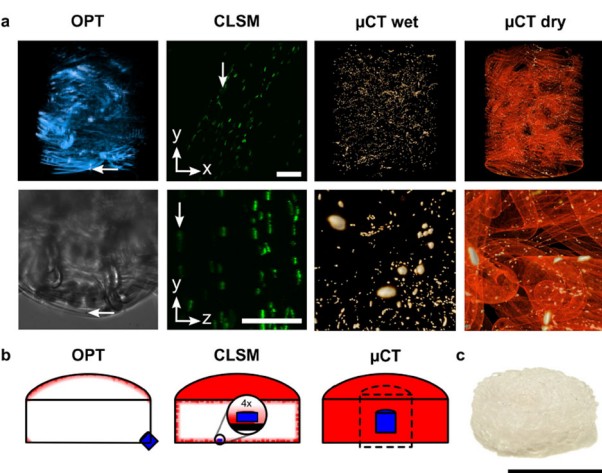

**Fig. 2 Visual comparison of nuclei in scaffolds.** Labeled nuclei in cell-cultured tissue-engineering scaffolds imaged with OPT, CLSM, and tube-source μCT (**a**). The reconstructed OPT data is shown above (voxel size 0.65 μm, more opaque and brighter color represent lower transmission intensity). One of the OPT bright-field transmission images is shown below. In both images, a nucleus is indicated by the arrows to clarify the image orientation. The CLSM data are shown from the aspect of the objective (the above image in *xy* pane) and in the side view close-up (the lower image in *yz* pane). In both images, a nucleus behind a PLA-fiber bundle is indicated by the arrows to clarify the intensity decay along the *z* axis. The CLSM voxel size is 0.55 μm, and scale bars are 100 μm. Whole μCT FOVs from the wet and dry imaging are shown in the above images (more opaque and brighter colors represent higher X-ray attenuation). Close-ups from the nuclei are shown below. The μCT voxel sizes are 2.2 μm and 1.1 μm for the wet and dry μCT imaging, respectively. Since 3D perspective was used to better represent the relatively large OPT and μCT FOVs, no scale bars are shown. As a reference, the dimensions of the OPT and μCT FOVs, indicated by the signal distributions in the above images, are 1.3 mm and 2.1 mm, respectively. The cross-sectional illustrations below each method visualize the part of the samples that can be examined by the respective method (**b**). By taking into account the image quality, imaging geometry, and the working distances of the objectives, areas that can be reconstructed are shown in red, and the acquired FOVs are shown in blue. The size of a μCT FOV that can be acquired with the 4x-objective is illustrated with the dashed lines (the dimensions are 5.5 mm; the voxel size would be 2.7 μm without CCD pixel binning). The real scaffold is shown in **c**. All the dimensions in **b** and **c** are adjusted in the same scale according to the central cross sections of the scaffolds, the common scale bar is 10 mm.

scaffolds (Fig. 2a). For this purpose, hASCs were seeded and lamin A/C antibody-silver labeled in fibronectin-coated polymer scaffolds composed of knitted and rolled polylactic acid[8] (PLA, ~5 mm high and 13 mm wide constructs, Fig. 2b, c). The diameter of the syringe used as a sample holder was 14.5 mm (imaging setup is shown in Supplementary Fig. 3). To illustrate the typical imaging problems associated with optically challenging samples, we captured the same silver signals in 3D using technically similar optical projection tomography (OPT) and more popular CLSM as reference methods (Fig. 2a, b). Similar to μCT imaging, the OPT data was acquired from the attenuated transmitted light, but in the CLSM imaging, the light reflecting back from the sample was used. In both optical imaging methods, a glycerin solution optimized for bright field imaging was used to minimize the refractive index mismatches (Supplementary Methods, Protocol 2). Since μCT is more sensitive to density variations, we wanted to demonstrate the imaging of the samples in both a water-rich state, a common state for various biomedical

samples, and after drying to maximize the contrast between the silver, scaffold, and background voids.

Intensive nuclear signals were seen in all the reconstructions. However, optical signals deteriorated even in the superficial volumes (Fig. 2a, b). All the nuclei and PLA fibers seen in the OPT reconstruction were badly distorted, especially below the immediate surface of the scaffold. In the CLSM reconstruction, most of the nuclear profiles were sharp in the *xy* plane, but were slightly smeared in the *yz* plane. Furthermore, clear intensity decay of the CLSM signals through a bundle of PLA fibers was seen along the *z*-axis. Unlike in the optical FOVs, µCT reproduced clear and intense nuclei from the surface (Supplementary Methods, Protocol 3 and Supplementary Movie 2) as well as from the middle of the sample (Fig. 2a and Supplementary Movie 3). The central µCT FOVs were about 7.25 mm deep along the imaging path from the surface of the sample-holding syringe. The absence of the attenuating water allowed larger fluxes of X-rays to pass through the sample. That is why we reduced the CCD pixel binning to decrease the voxel size from 2.2 µm to 1.1 µm for the dry-scaffold µCT imaging while keeping the imaging time relatively short (wet-sample µCT imaging without CCD pixel binning is demonstrated later). In the dry-scaffold µCT data, the PLA was also clearly visible due to its intermediate density between the silver and air. The demonstrated 2.1 mm wide and high µCT FOVs were large enough to capture thousands of cells. Details about the Fig. 2 image composition are given in Supplementary Note 4.

**Quantification of nuclear changes in scaffolds**. We also demonstrate how the tube-source µCT can be used to quantify cellular structures and the cell-function-related spatial changes in them. At the same time, we can examine in detail how a large amount of 3D quantitative data, a typical output of µCT imaging capable of capturing large FOVs, behaves in various steps of the in silico processing used (3DROQA, Supplementary Methods, Protocol 1). As an example subject phenomenon, we hypothesized that the rounding of nuclei caused by cytochalasin D-disturbed actin cytoskeleton[9] can be quantified from the µCT data. Antibody-silver labeled nuclei in the geometrical centers of the scaffolds are examined again, but this time untreated controls are statistically compared with cytochalasin D-exposed samples. The dry experiment with a voxel size of 1.1 µm is covered in Fig. 3, Supplementary Figs. 4–8, Supplementary Tables 1, 2. The wet experiment with a voxel size of 2.2 µm is covered in Supplementary Figs. 9–12, and Supplementary Table 3. The numbers of the µCT imaged scaffolds are 2 + 2 and 3 + 3 for dry and wet series, respectively. The developed 3DROQA includes a MATLAB code[10] that tabulates the spatial and intensity values for the statistical analyses executed in R (script available in Supplementary Data 1). The detailed quantitative data for each scaffold is available in Supplementary Data 2.

The quantitative data were processed as follows; the main focus is on the dry scaffolds. In short, the spatial quantification of the nuclei was achieved by first tuning the adaptive-segmentation threshold using superficial light microscopy as a reference (Supplementary Methods, Protocol 3). A few segmented example nuclei are shown in Fig. 3a and the superficial reference FOV is visualized in Supplementary Movie 2. To remove most of the small-object noise and incomplete objects cut by the edge of the FOV, and to represent the studied shapes by at least a decent number of voxels, 15 % of the acquired voxel particles in total (including hundreds of one to a few voxels objects Supplementary Fig. 4) were rejected in systematic data filtration (Supplementary Table 1). To calculate spatial measures and monitor data quality, the developed MATLAB code[10] fitted ellipsoids on to the rest of

the voxel particles using intermediate polygon particles (three example particle transformations shown in Fig. 3b–d). For example, ellipsoid-to-voxels volume ratios (distribution shown in Fig. 3e) were used to recognize data anomalies such as under-segmented, irregular aggregates of proximate nuclei that were known to be smoothed out by the ellipsoids, usually resulting in ratios over 1.2 (an example aggregate shown in Fig. 3f). However, only 11 % of the particles were rejected in the 0.8–1.2 band-pass filtration (Supplementary Table 1) as most of the ratios were tightly distributed near to a perfect one indicating useful quality data (Fig. 3e, more detailed representation in Supplementary Fig. 5). Furthermore, many of the initially rejected under-segmented aggregates were later successfully disintegrated in silico (Fig. 3f) while preserving the distinct nuclear features typical for the control and cytochalasin D-exposed samples (done only for the dry-sample data). The disintegration procedure is covered in Supplementary Fig. 6–8, and Supplementary Table 2. Further discussion and details about the overall experiment are available in Supplementary Note 5.

The quantitative results support the hypothesis that the tube-source µCT is a capable tool for quantifying microscopic cellular phenomena in 3D. The acquired data showed significant difference ($p$ values $<2.2 \times 10^{-16}$) in the studied shape measures toward rounder nuclei in the cytochalasin D-treated samples (Fig. 3g, the shape measures, sphericity, flatness, and elongation, approach one for a perfect sphere). The distinctiveness was also manifested in smaller average semi-inter quartile ranges (SIQR) than were the median differences between the control and cytochalasin D-exposed groups, excluding the elongation measurement in which the median difference was only about 44% of the average SIQR. The results of the shape measurements were compatible with the other observations as well. The significantly ($p$ value $<2.2 \times 10^{-16}$) increased attenuation intensity of the rounder nuclei indicated the compaction of the antigens and other nuclear matter, and maybe better labeling access into the loosened cell structure. Furthermore, the reduction and slight compensative protrusion of the longest and shortest nuclear axes (both $p$ values $<2.2 \times 10^{-16}$), respectively, were consistent with the flatness and elongation measurements based on the axial ratios. Thus, we were able to capture the cytoskeletal relaxation in the quantitative 3D data acquired from the centers of the relatively large scaffolds hard to image optically. Compatible results in similar trials were reproduced three times in total (dry, disintegrated dry, and wet data) as the shape measurements always indicated rounder nuclei in cytochalasin-D treated samples. Anomalies such as noise are expected to be present in the data to some extent. Further data enhancement is possible such as using conventional noise-reduction methods as discussed later in relation to Fig. 4. A further detailed discussion about the overall analysis and results is available in Supplementary Note 5. Details about the Fig. 3 image composition are given in Supplementary Note 4.

**Quantitative comparison of nuclei on flat substrates**. The tube-source µCT imaging was compared to optical quantitative 3D imaging using CLSM as a reference technique (Fig. 4). Silver and fluorescent labeling, based on the same antilamin A/C primary antibody, were used on hASC-cultured PLA-based flat-substrate samples that were optically easily accessible. The flat-substrate samples were left in a wet state, and the µCT and CLSM data were acquired with voxel sizes of 1.1 µm and 0.33 µm, respectively. 2 and 14 FOVs were acquired from 1 + 1 sample series with µCT and CLSM, respectively, data which were combined into distributions corresponding the imaging techniques. The

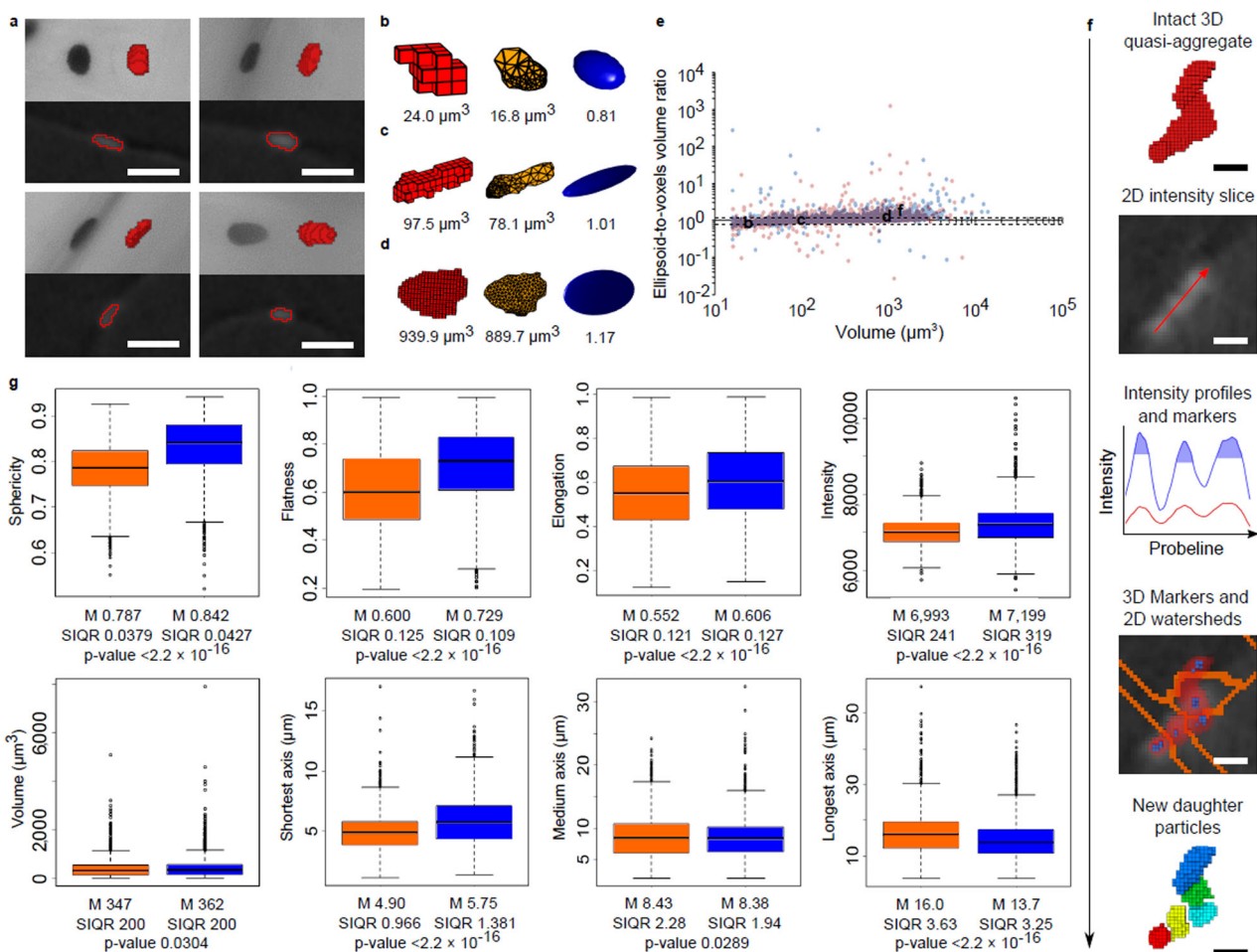

**Fig. 3 Quantification of nuclear changes in scaffolds.** Tube-source µCT data of dry-imaged nuclei in control and cytochalasin-D exposed scaffolds (1.1 µm voxel size). Four example segmentation-reference nuclei (**a**). The upper sections show the segmented voxel particles (red) next to the microscope images (grayscale). Tomographic slices from the nuclei are shown in the lower sections (grayscale) together with the segmentation interfaces (red frames). Scale bars are 25 µm. The found voxel particles (red) were transformed into ellipsoids (blue) through the polygon intermediates (yellow, three example cases **b**, **c**, and **d**). The ellipsoids are shown with the ellipsoid-to-voxels volume ratios. The example particles passed the 0.8–1.2 band-pass filter illustrated as dashed lines on the volume vs. ellipsoid-to-voxels volume ratio distribution (**e**, the red and blue dots refer to the control and cytochalasin-D exposed samples, respectively). The disintegration workflow (**f**) of an example under-segmented aggregate of proximate nuclei (uppermost red voxel particle). An intensity slice from the aggregate (grayscale) is shown with a probe line (red) measuring the intensity profile before and after contrast enhancement (red and blue graphs, respectively). The daughter particles were marked by thresholding the local intensity spikes (blue filled caps in the intensity profiles, not in scale). The found markers in 3D (blue) inside the aggregate (transparent red) are superimposed back on the intensity slice together with the found cross-sectional watershed interfaces (orange). In the end, five new daughter particles (multicolor) were obtained from the initially rejected aggregate. Scale bars are 10 µm. Statistics for the particles primarily accepted for the analysis without the disintegration, acquired from the 2 + 2 dry sample series (**g**). The boxes on the left (orange) represent the control group, and the boxes on the right (blue) represent the cytochalasin D-exposed group. The box-plots show the medians (thick center lines), the upper and lower quartiles (box edges), particles within ±1.5 × interquartile range of the upper and lower quartiles (whiskers), and the rest of the data (outlier circles). Medians (M), semi-interquartile ranges (SIQR), and the *p* values of Wilcoxon-Mann-Whitney tests are also shown. In total, 4360 and 4133 particles obtained from the control and cytochalasin D-exposed samples were analyzed, respectively.

quantitative data was processed using the 3DROQA (Supplementary Methods, Protocol 1). For the CLSM data, a proper segmentation threshold was chosen to fit the obtained voxel particles with the corresponding crisp nuclear fluorescence signals seen in the sharpest *xy* perspective (Fig. 4a). The µCT particles were obtained similarly, but using a light-microscope image of five reference nuclei on the studied sample (Fig. 4b, c, the reference-based adaptive segmentation principle described in detail in Supplementary Methods, Protocol 3). To establish a good reference, we verified that only true well-resolved nuclei were filtered into the CLSM data with good overall quality. For example, based on the manual investigation of all the FOVs, we used 120–2290 µm³ volume band-pass filtering on the CLSM

data. The used filtering led to the rejection of all the nonnuclear anomalies and only a few true nuclei (six in total, from which four were part of under-segmented aggregates), leaving 82 representative particles for the comparison. Following the more straightforward approach applied to the scaffold data (such as in Fig. 3), the µCT flat-substrate data was first investigated using only 13 µm³ high-pass filtration, leaving 312 particles for the initial comparison (Fig. 4d). How the µCT data was processed for further comparisons is discussed later. Detailed data flow through various steps for both the µCT and CLSM imaging are available in Supplementary Data 2.

In Fig. 4a, an example CLSM FOV is shown, with and without rejected data anomalies such as small-object noise. Similarly, in

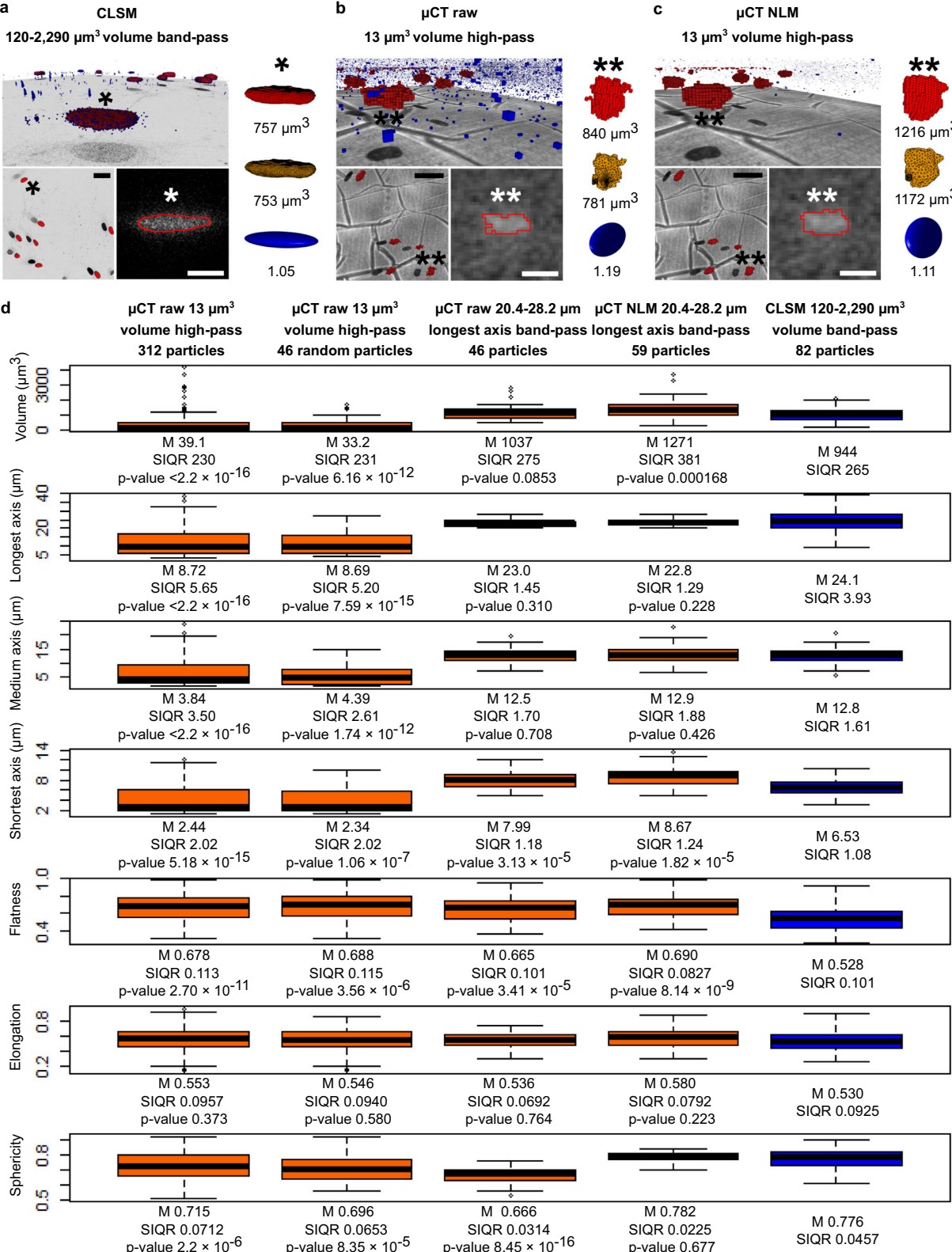

Fig. 4b, c, views on the μCT data without (raw) and with non-local-means (NLM) noise reduction are shown, respectively. As a basic noise-reduction method, the NLM was demonstrated to compensate for the noise in the ambitious wet-state μCT imaging without the CCD pixel binning. Without and with the NLM enhancement, the segmentation resulted in satisfying overall fits of the μCT voxel particles with the light-microscope reference.

From the five μCT reference nuclei in the raw and NLM data, 4/5 and 5/5 passed through the 0.8–1.2 ellipsoid-to-voxels volume-ratio filter, respectively, used as a data-quality indicator. The routinely fitted ellipsoids enhanced the manifestation of the smooth morphology of the true nuclei. The additional NML enhancement made the voxel and polygon particles smoother and more solid as well.

**Fig. 4 Quantitative comparison of nuclei on flat substrates.** 3D quantitative comparison of wet-state nuclei on flat substrates imaged with tube-source µCT and CLSM (voxel sizes of 1.1 µm and 0.33 µm, respectively). An example CLSM FOV is shown in **a**. A landscape view along the sample surface is provided in the upper section, including the voxel particles used for the comparison (red), and the rejected data anomalies (blue, no scalebar due to the 3D perspective). Below the voxel particles, a CLSM cross-section is shown in *xy* perspective (darker tones represent higher signal intensities). In the lower-left section, the accepted voxel particles are shown in *xy* perspective with the cross-section (the scale bar is 50 µm). In the lower-right section, a cross-section perpendicular to the substrate surface (brighter tones represent higher signal intensities) together with the segmentation interface (red frame) are shown (the scale bar is 10 µm). On the right, an example particle transformation is shown: a voxel particle (red) was transformed into ellipsoid (blue, with the ellipsoid-to-voxels volume ratio) through the polygon intermediate (yellow). An example nucleus followed in the image sections is marked with the asterisk (*). Following the previous presentation, µCT data without and with NLM noise reduction is shown in **b** and **c**, respectively. In the upper and lower-left sections, the light-microscope image of the five segmentation-reference nuclei is shown (grayscale). Statistics (**d**) for the particles acquired from the 1 + 1 PLA-based flat-substrate sample series, from which 2 and 14 FOVs for combined data were acquired with µCT and CLSM, respectively. The µCT data processed in different ways (text labels) for the statistical analyses are shown in the boxes on the left (orange). The CLSM data, only including manually verified well-resolved nuclei, is shown in the boxes on the right (blue). The box-plots show the medians (thick center lines), the upper and lower quartiles (box edges), particles within ±1.5 × interquartile range of the upper and lower quartiles (whiskers), and the rest of the data (outlier circles). Medians (M), semi-interquartile ranges (SIQR), and the *p* values of Wilcoxon-Mann-Whitney tests for the µCT vs. CLSM comparisons are also shown.

The quantitative similarities and differences between the µCT and CLSM particle distributions can be seen in Fig. 4d. Excluding the elongation shape value, which was the most stable measurement along the whole investigative path (reading the µCT boxplots from left to right, the *p* values varied between 0.223 and 0.764), the initial raw µCT data was rather dissimilar when compared to the CLSM reference data. To decrease the effect of the biological variation, and to reduce the number of the expected anomalies (such as the non-nuclear background spots also rejected from the CLSM data shown in Fig. 4a), the following was done to focus the investigation on the particles sharing similar lengths. Knowing that the CLSM data only presents true well-resolved nuclei, and that the longest axes are aligned with the sharpest *xy* plane in the CLSM imaging, and that the error sources affect proportionally less the largest dimensions, the µCT data was filtered through the 20.4–28.2 µm longest axis band-pass (Fig. 4d). The limits correspond to the lower and upper quartiles of the CLSM data. To verify that the lower particle number does not cause remarkable changes in the statistics, we first investigated a randomly picked group of raw µCT particles, in which the number of particles corresponds to the longest axis band-pass filtered distribution (46 out of the 312 initial particles). No remarkable changes with the initial group of raw µCT particles were observed (similar distributions and similar *p* values when compared to the CLSM data). However, in correlation with the squeezed long-axis distribution of the µCT data, the medium axis and volume measurements between the two imaging methods lost their strong statistically significant differences (*p* values changed from $1.74 \times 10^{-12}$ to 0.708, and from $6.16 \times 10^{-12}$ to 0.0853, respectively). Also, the median differences (0.3 µm and 93 µm³) fitted well within the average SIQRs (1.66 µm and 270 µm³ for medium axis and volume, respectively). Thus, the nuclei sampled into the µCT and CLSM data seem to share similar dimensions aligned on the cell-cultured surfaces, seen from the most reliable *xy* perspective, and the volumes are alike as well (median volume difference of 9.85%, larger for the µCT particles as expected due to the coarser voxel size and imaging artifacts).

After applying the NLM enhancement, to clean the noise-induced surface irregularities of the µCT particles (Fig. 4b, c), the sphericity distribution also aligned with the CLSM data with low statistical significance (*p* value 0.677, Fig. 4d). The median difference (0.006) also fitted well within the average SIQR (0.0341). However, as the NLM seemed to have a filling effect on the surface irregularities (Fig. 4b, c), the median volume difference to the CLSM data increased to 34.6%, and the differences became statistically significant (*p* value 0.000168, Fig. 4d). After the NLM image enhancement, the volume median difference and average SIQR were 327 µm³ and 323 µm³,

respectively. Thus, 4/7 of the compared measurements had statistically insignificant differences, and the distributions were aligned, but the matching measurements varied depending on the use of the NLM noise reduction applied to the µCT data. A further detailed discussion about the comparison of the imaging methods is available in Supplementary Note 5. Details about the Fig. 4 image composition are given in Supplementary Note 4.

## Discussion

We have shown that a commercial tube-source µCT can obtain visually and quantitatively representative data of cellular structures and phenomena with good 3D coverage. The µCT imaging was demonstrated as a part of the methodological pipeline combining cell-structure specific labeling, X-ray imaging, and the newly developed in silico processing workflow (3DROQA, Supplementary Methods, Protocol 1). Thousands of cells were captured in the µCT FOVs providing vast amounts of spatial data from which we were able to quantify even a subcellular phenomenon (Fig. 3, Supplementary Figs. 8, 9). The acquisition of a variety of cellular structures was demonstrated with the image accuracy greatly resembling that of the used optical reference methods (visual comparison of different cell structures in Fig. 1, quantitative comparison of nuclei in Fig. 4). The used antibody-silver label enabled the detection of antigens regardless of the optical properties or the attenuating water content of the samples, while other materials are simultaneously distinguishable if they possess appropriate density variations. Our results highlight that relatively large cell-cultured samples can be investigated thoroughly with a tube-source µCT, while the used optical reference methods enabled only superficial inspection (Fig. 2).

The 3D quantification was demonstrated with cylindrical µCT FOVs having both width and height of 2.1 mm in dry and wet cell-cultured scaffolds, with voxel sizes of 1.1 µm and 2.2 µm, respectively. Wet-sample µCT imaging without CCD pixel binning, resulting in a voxel size of 1.1 µm, is demonstrated in Fig. 4. We would like to note that µCT FOVs having even bigger dimensions of 5.5 mm, owning a microscopic voxel size of 2.7 µm, are also possible to acquire with the µCT device by using a less-magnifying 4x objective (only the use of the 10x objective was demonstrated in this paper). The dimensions of the µCT FOVs and how they relate to the examined clinically-relevant sized tissue-engineering scaffolds are illustrated in Fig. 2b. Supplementary Movies 2 and 3 are also useful to better perceive how large the demonstrated FOVs are in relation to the dimensions of the nuclei. The 3D microscopic accuracy was demonstrated with an X-ray transmission pathway in which samples with a maximum width of 14.5 mm can be fitted for the free positioning of the FOV (including the used sample-holding syringe,

Supplementary Fig. 3). Samples having a total width of about 26 mm also fit into the transmission pathway if the region-of-interest is positioned in the center of the samples to avoid the collision between the rotating sample and the objective.

The demonstrated antibody-silver labeling is suitable for samples owning open structures where the labeling components are easy to introduce. These kinds of samples include fibrous and porous tissue-engineering scaffolds and organ-on-a-chip constructs with open channels and cavities. To modify the µCT-based methodological pipeline for studying more confluent 3D samples, semi-specific labeling based on more diffusive agents could be used to substitute the antibodies, such as using hematein lead(II) complex for labeling nuclei in tissues[11]. A short introduction to available contrasting methods for various samples and target structures is available in Supplementary Note 2. In addition to nuclei, other interesting cellular structures for quantification could be the components of the cytoskeleton, cell-cell junctions (Fig. 1), whole cells themselves, or super-cellular structures such as cell aggregates or extra-cellular components. As an example, the size of the cells could be used to track the differentiation process. The size can also vary in response to physiological stimuli or imply about the metabolic state of the cells[12]. To better relate the size of the quantified hASC nuclei to other cell types, in the review article written by Ginzberg et al.[12], different cell types are conveniently illustrated for the size comparison. For example, the µCT imaging of the nuclei of adipocytes, fibroblasts, and keratinocytes, or whole pancreatic or hepatocytic cells, could result in data analogous to the hASC nuclei presented in this paper. In addition to the hASC, µCT imaging of the Caco-2 epithelial cells was demonstrated in Fig. 1.

The antibody-silver signals were extracted for the quantification using the reference-based adaptive segmentation (a few segmented example nuclei are shown in Figs. 3a, 4b, c, and Supplementary Fig. 10). In Methods and Supplementary Methods, Protocol 3, we explain in detail how to adjust the parameters for a successful segmentation using conventional light microscopy as a reference and how to take into account the spectral variation of the X-rays interacting with the subject material. The X-ray attenuation-based signals were strong enough to be extracted from both the dry (Fig. 3) and wet-state data acquired with and without the CCD pixel binning (Supplementary Fig. 10 and Fig. 4b, c, respectively). In addition to the CCD pixel binning, it was possible to reduce the noise with data-processing methods such as volume filters and NLM. On the other hand, attenuating water helps to reduce the X-ray beam-hardening artifacts, effect visible in the superficial FOV obtained from the dry scaffold (Supplementary Methods, Protocol 3 and Supplementary Movie 2).

To assess the achieved µCT resolution, we need to look at the two limit-reaching cases. First, despite the image accuracy that greatly resembles the used optical reference methods, we did not manage to reproduce the curly ZO-1 features in the µCT data having a voxel size of 1.2 µm (Fig. 1). However, the resolution of larger objects much less than 2.4 µm apart is possible in silico despite the resolution limit set by the Nyquist sampling theorem and the X-ray image artifacts such as penumbra. For example, we were able to separate the proximate nuclei in the initially under-segmented aggregates, one case shown in Fig. 3f (voxel size 1.1 µm). Taking into account the principle of how the used 3D marker-based watershed separation works, it can be used to resolve multi-voxel objects even in full contact, assuming that the markers are correctly placed one per individual object. Many characteristic features of the control and cytoskeleton-relaxed nuclei (Fig. 3g) were preserved in the data obtained from the in silico disintegrated aggregates (Supplementary Fig. 8). Thus, in silico processing can be used to resolve certain types of proximate

objects which interfaces are initially smeared due to X-ray imaging artifacts. However, the micro-focus tube-source X-ray imaging is not suitable for studying ultrastructures that can fully fit within the micro-scale voxels. To study submicron structures, nano-CT devices and synchrotron technology are more suitable. A short introduction to various X-ray imaging technologies is available in Supplementary Note 3.

Considering the detailed quantitative comparison to the CLSM data in 3D, the µCT was able to produce good representations of the nuclei sharing many similarities with the reference data (Fig. 4). The µCT data was coarser to some extent as expected due to the larger voxel size and the characteristic X-ray tube-source image quality. The µCT imaging enables the acquisition of within-technique comparable quantitative data, for example, for studying relative spatial changes in control and experimental cell structures (such as demonstrated in Fig. 3). If more universal, cross-technique comparable quantitative data is needed, using a reference 3D-imaging technique for finding the proper processing settings for the µCT data could be needed. The other option is to use known reference subjects, such as standardized micro-particles, which features are known, and their X-ray characteristics match with actual subject details under the investigation. To assess the accuracy of our tube-source µCT method, known materials such as microparticles could have been used. However, the comparison of the µCT method with CLSM on single-layer cell-nuclei imaging provides an excellent validation method as it highlights the entire labeling, imaging, and image-processing chain of the true cell structures themselves.

To analyze cellular structures with microscopic accuracy and good 3D coverage, we have demonstrated the potential and studied the characteristics of tube-source µCT. We showed how the commercially available X-ray imaging technology could be used to study optically challenging cell-cultured constructs, both visually and quantitatively. We made visual and quantitative comparisons to the optical imaging reference techniques, and provide comprehensive and pragmatic backgrounds with explanations to adapt the technique further. With the detailed know-how disclosed in this paper, a variety of optically challenging cell-cultured constructs, such as samples including porous, fibrous, rough-surfaced, composite, and pigmented materials, can now be imaged with microscopic accuracy and good 3D coverage for representative cellular visualization and quantification.

## Methods

**Preparation of PLA samples**. To prepare the scaffolds, medical-grade poly(L/D) lactide 96/4 copolymer (PLA, Purasorb PLD 9620, Purac BU Biomaterials) was melt spun into 4-filament fiber using a microextruder (Gimac di Maccagnan Giorgio) and a laboratory-scale fiber spinning line (Fourné Polymertechnik GmbH). Before melt spinning, the material was dried in vacuum at 100 °C for 16 h. The scaffolds were knitted and rolled as before[8]. The scaffolds were gamma sterilized with 25 kGy radiation dose.

To prepare the PLA-based flat-substrate samples, the following was done. The medical-grade PLA was processed into 0.75 mm thick sheet using a form press machine (ZB110, Nike Hydraulics). 15 mm diameter round pieces were punched from the sheet. To ease the manipulation in 24-well plate, and to fit the other PLA sheet into the sample-holding syringe, ~2 mm pieces were cut from the opposite sides of each PLA sheet. For sterilization, the PLA sheets were washed in 99.5% ethanol (Etax Aa, Altia Oyj) in an ultrasonic cleaner (M12, FinnSonic) 2 times. Each washing cycle lasted for 2 min, and the ethanol was changed between the cycles. The sheets were sealed into ethanol sterilized plastic pouch in a laminar flow cabinet using an aseptic technique.

**Cell culturing**. hASCs were isolated from an adipose tissue sample obtained surgically from a 50-year old female donor. Permission to conduct the procedure was given by the Ethics Committee of Pirkanmaa Hospital District, Tampere, Finland (R03058, R15161). The donor gave written informed consent for the use of the adipose tissue sample for research purposes.

The isolation of hASCs from the adipose tissue sample was accomplished by a mechanical and enzymatic procedure described previously[13,14]. The isolated hASCs were maintained in T-75 polystyrene flasks (Nunc, Thermo Fisher Scientific) in

DMEM/F-12 (Life Technologies, Thermo Fisher Scientific) supplemented with 5% HS (PAA Laboratories GE Healthcare), 1% L-glutamine (GlutaMAX I, Life Technologies, Thermo Fisher Scientific) and 1% antibiotics (100 U/ml penicillin and 0.1 mg/ml streptomycin; Lonza). The cells were routinely tested for mycoplasma contamination with negative results.

Prior to the cell culture, the PLA-based samples (the scaffolds and sheets) were coated by incubating them in 25 ug/ml fibronectin in phosphate-buffered saline (PBS, BE17–515 F, BioWhittaker®, Lonza) solution for 48 h at + 37 °C. The used fibronectin was isolated from human plasma as described earlier[15]. Following the coating, 100 000 cells were applied to each PLA scaffold in a volume of 100 µl. After 3 h attachment, the culturing volume was increased to fully cover the scaffolds. After 4 days, half of the samples were treated with 5 µg/ml cytochalasin D (#C2618, Sigma) in a volume of 1 ml cell culture medium in a 24-well plate format for 1 h at + 37 °C. The cytochalasin D was primarily dissolved in dimethyl sulfoxide, and hence 1 µl/ml of dimethyl sulfoxide (D2650, Sigma-Aldrich) was applied for both control scaffolds and the ones being exposed to the cytochalasin D. Cytochalasin D treatment was stopped by washing all the samples four times with PBS. To study the cells on the well-plate-based flat-substrate samples, approximately 1000 cell/cm² were similarly cultured in well plates (Nunc, Thermo Fisher Scientific) for 4 days without the fibronectin and cytochalasin steps.

Caco-2 cells (ATCC HTB-37) were cultured in Minimum Essential Medium (#41090028, Thermo Fischer) with 10 % FBS (#10270–106, Thermo Fischer), sodium pyruvate (#11360039, Thermo Fischer), sodium bicarbonate (#25080060, Thermo Fischer), non-essential amino acids (#BE13–114E, Lonza), and Penicillin-Streptomycin antibiotic (#15140122, Thermo Fischer). The cells were routinely tested for mycoplasma contamination with negative results.

**Cell fixation.** All the cells were fixed using principally the same conventional 40 mg/ml formaldehyde in PBS solution. All the so-called flat-substrate cells were fixed using paraformaldehyde (158127, Sigma-Aldrich), the fixative was prepared as follows. 40 g of solid paraformaldehyde was dissolved in 900 ml water heated to 65 °C and mixed for 45 min. A few drops of 1 M NaOH was added until the solution became clear. The solution was cooled and mixed with 100 ml of 10X PBS, and pH was titrated to 7.3 using 1 M NaOH or HCl. Finally, the fixative was filtered and stored in −20 °C aliquots for later use. The cells cultured in the PLA scaffolds were fixed using more convenient 1/10 dilution of commercial 10–15% methanol-stabilized 37 wt. % formalin (F1635, Sigma-Aldrich) mixed with 1/10 10X PBS and 8/10 water. Although two variations of the principally same fixative were used, the same fixative recipe was used for all the within-experiment cells to ensure comparability.

**Fluorescent labeling on well-plate-based flat substrates.** To make the visual comparisons, the nuclei and actin cytoskeleton of the well-plate-cultured hASCs were enhanced by DAPI (Molecular Probes, Thermo Fisher Scientific) and tetra-methylrhodamine B isothiocyanate conjugated phalloidin (P1951, Sigma-Aldrich), respectively. Phalloidin was also used for the Caco-2 cells. The culturing medium of the cells was removed, and the cells were rinsed and washed 3 times for 4 min with PBS at room temperature with mild lateral agitation (referred to as "washing" from now on). The cells were fixed for 10 min, after which the cells were washed. Cells were permeabilized with 0.1% Triton X-100 (T8787, Sigma-Aldrich) in PBS for 10 min. After washing, the cells were blocked with 10 mg/ml bovine serum albumin (BSA, A7906, Sigma-Aldrich) in PBS for 30 min. To fluorescence label the nuclei and actin cytoskeleton, the cells were kept in the dark and subjected to either 0.5 µg/ml DAPI in PBS for 5 min, or 5 µg/ml phalloidin in 10 mg/ml BSA in PBS for 45 min, respectively. Then, the samples were rinsed and washed once with PBS for 4 min and twice with milli-Q-water (18.2 MΩ·cm resistivity at +25 °C). The mitochondria of the well plate-cultured hASCs were labeled with MitoTracker® Red CMXRos (M-75112, Life Technologies, Thermo Fisher Scientific). The stock solution stored at −20 °C was thawed and diluted in the hASC culturing medium to a 20 nM working concentration. After removing the previous medium, the cells were subjected to the prewarmed labeling solution for 30 min in a cell incubator. The labeling medium was discarded and replaced to a fresh medium, in which the cells were fluorescence-imaged.

**Antibody labeling.** Horseradish peroxidase (HRP) functionalized secondary antibody was used to produce intense, non-diffusible, and well preserving silver precipitates[16–18] for µCT imaging. To avoid day-to-day variance affecting the results, all the quantitatively compared samples were treated parallel in the same sample series. To ensure stable conditions and avoid drying during long-lasting steps, sample-holding well plates were sealed with Parafilm, and some unused wells were filled with PBS to retain the moisture equilibria under the lids. All the following steps were made by keeping the samples in gentle lateral agitation at room temperature, and enough liquids were used to fully cover the samples.

Cell samples were briefly washed with PBS. The samples were fixed for 10 min. After washing, the cells were permeabilized with 0.1% Triton X-100 in PBS for 10 min. After washing, the samples were blocked by subjecting them to the ready-to-use blocking solution included in the HRP functionalized secondary antibody labeling kit (MP-7402, Vector Laboratories) for 1 h. The monoclonal mouse IgG

primary antibodies were diluted in 5 mg/ml BSA (A7906, Sigma-Aldrich) in PBS solution. The following dilutions, as recommended by the manufacturers for conventional antibody labeling, were used: 1:100 anti-β-actin (C4 sc-47778, Santa Cruz Biotechnology); 1:200 antilamin (A + C [131C3] ab8984, Abcam); 1:1000 anti-ATP5α (ab14748, Abcam); 1:100 anti-ZO-1-1A12 (33–9100, Invitrogen). After removing the blocking solution, the samples were kept in the primary antibody solutions for 1 h. To keep the non-specific protein binding sites still crowded, unbound primary antibodies were washed away with 5 mg/ml BSA in PBS solution. The mouse-specific ready-to-use HRP secondary antibody was applied for 1 h. Finally, unbound secondary antibodies were washed away first by PBS and then with milli-Q-water. It is important to use such highly pure water to avoid the secondary effects of the silver generation reagents, and the formation of X-ray attenuating salt precipitates in the following steps.

As a prime control test during the labeling of the scaffolds, we made a parallel no-primary antibody experiment to unveil possible endogenous peroxidase activity, nonspecific secondary antibody binding, silver reaction background, salt precipitates, or insufficient sample protection against impurities such as atmospheric dust particles. According to the control experiments and high signal-to-noise ratios in the labeled samples, no significant endogenous peroxidase activity was observed during our experiments (no-primary-antibody scaffold control presented in Supplementary Movie 4). Thus, conventional but harsh peroxidase blocking was excluded to guarantee the maximum immunosensitivity and signal intensity of the studied antigens[19].

Following the protocol described above, 1:200 diluted A488 goat anti-mouse (IgG (H+L) highly cross-adsorbed, Invitrogen) secondary antibody, and 1:20 diluted serum (G9023, Sigma) in PBS were used to fluorescent label the nuclei on the PLA-based flat-substrate sample used for the quantitative comparisons. The steps, including the fluorescent label, were done in lowered lab lightning. At the end of the labeling, the sample was not washed with milli-Q-water but left in PBS for CLSM imaging, to preserve the cells in physiological circumstances.

**Silver generation.** A commercial silver generation kit (#6010 EnzMet™ for General Research Applications, Nanoprobes) was used for generating HRP-localized silver particles. To avoid possible photochemical reduction of silver, all the silver generation steps were conducted in decreased laboratory lightning. Thus, to avoid the secondary effects of light, the silver reactions were not observed with a light microscope, although 610 nm long-passed red light could be used as discussed later (Supplementary Movie 1 in which the appropriate signal distribution during 15 min silver generation was recorded). The recommended maximum silver generation time was used to maximize the signal intensities, compensate possibly varying antigen expressions, and chemically equalize even the deepest parts of the complex scaffold structures. The cold stored reagents were thermally stabilized at room temperature prior to their use.

By using gentle suction, water was removed from the well plates, and the samples were subjected to reagent A. The simple stepwise protocol was further continued in accordance with the kit manual until the final 15 min silver generation step was accomplished. Due to the limited volume of the used wells to hold the final reagent mixtures, the PLA scaffolds were not fully covered at the beginning. That is why the samples were constantly but gently rinsed with the reaction solution until full liquid coverage was achieved. Otherwise, the samples were kept in mild lateral agitation. Rinsing also helped with mixing and introducing the reaction solution deep inside the voids of the scaffold. The silver generation was stopped by rinsing the samples multiple times with milli-Q-water to remove the reaction substrates. The samples were stored at +4 °C and protected from light and drying.

To verify the proper distribution of the silver signal, an additional silver-generation experiment with antilamin labeled hASC on the PLA-based flat-substrate sample was video recorded with silver-safe red-light microscopy (Supplementary Movie 1). In a prior in vitro test, we verified that the used 610 nm long-pass (FGL610M, Thorlabs) filtered red light did not produce visible silver precipitate or Tyndall effect observed with a red laser pointer during 20 min exposure of the silver-generation reaction mixture. The result was similar to the corresponding control solution kept in lowered ambient lab light, excluding intermittent short observations of the Tyndall effect with the laser pointer.

**Light and fluorescent microscope imaging.** The cells in the well plates were observed directly with light (EVOS Fl Auto Cell Imaging, Life Technologies, Thermo Fisher Scientific) and fluorescence (IX51, Olympus) microscopes and included 20x air objectives (LPlanFL PH2 20x/0.40, Thermofisher and LUCPlanFL N 20x/0.45, Olympus, respectively). Appropriate source intensities, exposure times, and gains were set for acquiring representative, low noise, and well but not over-exposed images. To observe the fluorescent signals of DAPI and phalloidin, excitation wavelengths of 360–370 nm and 540–550 nm, and emission windows of 420–460 nm and 575–625 nm were used, respectively. MitoTracker Red was studied with the same wavelengths as the phalloidin labeled samples. The silver-generation in Supplementary Movie 1 was recorded with Axio Vert.A1 light microscope equipped with EC EPIPLAN 20x/0.4 objective and Axiocam 105 color camera (Zeiss).

**Sample preparation for CLSM, OPT, and µCT imaging**. For the visual and quantitative comparisons, well-plate-based and PLA-based flat-substrate samples were used, respectively. The light-microscope and fluorescence imaging were done directly on the intact well plates. To acquire the labeled well-plate-based flat-substrate samples for the µCT and CLSM imaging, the bottom pieces of the well plates were detached (Supplementary Methods, Protocol 4), as the whole plates did not fit in the µCT imaging setup (Supplementary Fig. 3). Turning the bottom pieces upside down for the CLSM imaging, it was possible to examine the cells through optically good coverslips glued as bottoms in the otherwise plastic dish; the same approach was used with the PLA-based flat-substrate samples. Glassy materials were not used as culturing substrate due to their inherent X-ray attenuating properties. Applicable bottom pieces were attached either to the µCT sample chuck and let to dry, or were imaged by the CLSM in milli-Q-water. To protect the samples from airborne dust particles and gently high-pass filter the X-rays, the µCT sample chuck was surrounded by a 5 ml polypropylene syringe (SS*05SE1, Terumo) with an outer diameter and wall thickness of 14.5 mm and 0.8 mm, respectively (the so-called "sample-holding syringe").

To µCT image the PLA scaffolds and PLA-based flat-substrate samples, they were laid in a polypropylene syringe similar to the one mentioned above, filled with milli-Q-water (Supplementary Fig. 3). The finger flanges of the syringe were removed to allow the close positioning of the detector. The tip of the syringe was attached to the sample chuck while the openings were covered with Parafilm. After the wet imaging, further µCT-imaged scaffolds were removed from the milli-Q-water and dried under the draft of a fume hood for at least 24 h in a loosely covered well plate at room temperature. After drying, the samples were returned into the empty syringe for the dry µCT imaging. All the acquired µCT FOVs were at least 2 mm above the metallic sample chuck to decrease the number of scattering photons. To avoid motion artifacts, the sample-holding syringe was knocked gently and left to stabilize mechanically on the sample platform overnight with a couple of hours under X-ray exposure.

After the wet µCT imaging, one of the control PLA scaffolds was immersed in a bright field optimized 88% glycerin solution for optical 3D imaging (Supplementary Methods, Protocol 2). Water was removed from the scaffold by draining multiple milliliters of the glycerin onto the scaffold, and finally immersing the sample into fresh liquid. The bottom of the immersed scaffold was first imaged by the CLSM in the previously described plastic dish with the glass bottom. To maximize the sample stability in the subsequent OPT imaging, the scaffold was then sewn with a piece of line into a correctly cut plastic tube. The tube acted as an adapter between the sample and the rotating sample rod attached to the manual x-y-stage in the OPT setup.

**CLSM imaging**. CLSM (LSM780, Zeiss) with 40x/1.2 (C-Apochromat, Zeiss) and 25x/0.8 (LD LCI Plan-Apochromat, Zeiss) objectives were used to image the well-plate-based flat-substrate and the PLA-based samples (scaffolds and sheets), respectively. Immersol W (2010) and Immersol G immersion liquids (Zeiss) were used with samples immersed in water and glycerin mixture, respectively. ZEN 2011 SP3 (black edition, 8.1.0.484, Zeiss) software was used. To acquire reflection images for the visual comparisons, illumination wavelength of 594 nm was used. To acquire the fluorescence images for the quantitative comparisons, excitation wavelength of 488 nm was used. Source intensities, exposure times, and detector gains were adjusted to collect low-noise, well-exposed images. The flat substrate samples immersed in milli-Q-water were scanned by acquiring 353.9 µm × 353.9 µm images with a pixel size of 0.35 µm in the $xy$ plane. The scaffolds were immersed in a glycerin solution optimized for bright field imaging (Supplementary Methods, Protocol 2) and scanned by acquiring 566.2 µm × 566.2 µm images with a pixel size of 0.55 µm in the $xy$ plane. $z$ scanning intervals of 0.196 µm and 0.549 µm were chosen for the flat substrate and the scaffold samples, respectively. The acquired $z$ stacks were thick enough so that all the target features were scanned thoroughly from blur-to-blur seen in the transmittance view. 4- and 2-line averaging was used to scan the flat substrate and scaffold samples, respectively. The acquired reflection data were reconstructed by using the 3D viewing mode in the ZEN 2011 software. To acquire the quantitative data for Fig. 4, a small stainless nut was laid on the PLA-substrate to maximize the sample stabilization.

**OPT imaging and reconstruction**. The x-y-stage (7T167–25XY, Standa) was used to adjust the center of rotation of the sample. The x-y-stage was connected to a sample positioning module comprising a motorized rotation stage and an x-y-z motorized linear stage (8MR190–2–28 & 8MT167–25LS-XYZ, Standa). The module was used for sample alignment and rotation.

The sample was bright field transilluminated with a white LED (SL4301A-WHI IC, Advanced Illumination Inc.) and telecentric backlight illuminator (Techspec 62–760, Edmund Optics). The detection path consisted of a 10x infinity-corrected, long working distance objective having a numerical aperture of 0.28 (EO M Plan Apo 10x 59–877, Edmund Optics), and a tube lens (U-TLU-1–2, Olympus). The images were collected with a 2048 x 2048 pixel sCMOS camera (C11440–22CU, Hamamatsu Photonics). Thus, together with the used optics, the physical pixel size of 6.5 µm × 6.5 µm gave a virtual pixel size of 0.65 µm × 0.65 µm. The system and imaging were controlled via LabView (LabView 2013, National Instruments).

OPT[20] images were acquired as 16-bit photon counts from 400 evenly distributed projection angles around the sample in full 360 degrees. The exposure time of 14 ms was found to give well-exposed images with minimal saturation.

To acquire the axis of center of rotation, two preliminary slices were reconstructed by trying different horizontal offsets[21]. The slices were chosen to include the lowest nuclei possible in the FOV, farthest away from the thicker upper parts of the hanging scaffold (Fig. 2a). The axis was chosen manually with visual assessment and metrics, as described earlier[21]. The chosen axis was a straight horizontal line with 29-pixel offset; subpixel deviations were not considered. OPT reconstruction is a computation of inverse Radon transform with an offset correction. A ramp filter with Hann window was used to correct illumination irregularities[22].

**µCT setup and image acquisition**. Tomographic data were acquired by using commercial tube-source µCT (Xradia MicroXCT-400, Carl Zeiss), including a microfocus X-ray tube source (L8121–01, Hamamatsu Photonics) and a 2048 x 2048 CCD camera with a pixel size of 13.5 µm (iKon-L936, Andor Technology Ltd). A scintillator-equipped 10x objective (other options also available such as 4x), integrated into the µCT system by the manufacturer, was used to acquire all the tomographic data. The µCT was operated using XMController software provided by the system manufacturer. To achieve sufficient voxel sizes, good exposure, effective attenuation contrast, diminutive image artifacts such as penumbra, and keeping the imaging time reasonable (as a reference, not exceeding generally accepted multiple days needed for clarifying tissues, for example), the following imaging parameters were chosen.

The object-to-detector distance was set as small as possible without colliding the detector with the rotating samples, meaning 11 mm and 15 mm for the Fig. 1 well-plate-based flat-substrate samples and rest of the imaging, respectively. The 15 mm object-to-detector distance enables the complete coverage of the scaffolds, although the examination focused on the geometrical centers of the samples. In Fig. 1, the smallest possible source power of 4 W was used to minimize the focal spot size down to 5 µm, but for the rest of the imaging, the power was raised to 10 W resulting in 7 µm focal spot. Altogether with the used 60 mm source-to-object distance, these arrangements enabled non-rounded voxel sizes of 1.16317 µm, 1.10114 µm, 2.20227 µm, and 1.09223 µm, for the well-plate-based flat-substrate, dry and wet scaffold, and PLA-based flat-substrate samples, respectively. The average penumbras[6] were 0.9 µm and 1.8 µm for the well-plate-based flat-substrate samples and rest of the data, respectively. To compensate the attenuating water and resulting noise in the wet scaffold imaging, the use of $2 \times 2$ CCD pixel binning was demonstrated to increase the exposure with the larger voxel size. In Fig. 4 we demonstrated wet-state imaging without CCD pixel binning. The small 1.10114 µm vs 1.09223 µm deviation in the voxel sizes originates from the routine maintenance and calibration of the µCT device between the experiments.

The X-ray images were acquired from 1600 evenly distributed angles around the samples. In addition to increasing the signal-to-noise ratio and contrast in the reconstructions[23], the high number of projections was favored to keep the geometrical image accuracy high. By considering the size of the FOVs (2.1 mm wide in Figs. 2, 3, and 4) and the voxel sizes, the angular sampling farthest away from the center of rotation was theoretically too low to utilize the available geometrical resolution in the non-CCD-pixel-binned imaging fully, but not in the wet-scaffold imaging. This was considered an insignificant source of image errors due to the lack of straight edges in the studied scaffolds, but it could affect more the edge most nuclei in Fig. 4[5]. As an example visualization in Fig. 2, the 2.1 mm wide and high FOV can be changed to even larger 5.5 mm FOV with 2.7 µm voxel size by changing the used 10x objective to 4x, but this kind of imaging was not investigated in practice. As the exposure would increase due to the decreasing magnification in 4x imaging, possible geometrical inaccuracies could be compensated with a larger number of projections with shorter exposure time.

Exposure times of 50, 22, 8, and 30 s were used for the well-plate-based flat-substrate, dry and wet scaffold, and PLA-based flat-substrate samples, respectively. In addition to exposure times, 80 kV acceleration voltage was chosen to achieve >5000 count intensity in the major parts of the FOV (Supplementary Fig. 13), a recommended intensity minimum given by the µCT manufacturer. The suitability of the chosen acceleration voltage was theoretically verified by the aid of an online X-ray spectrum simulator[24]. In a theoretical 80 kV spectrum reaching the samples, most of the photon energies slightly exceed the K-edge energy of silver, which is 25.5 keV[25]; the most probable photon energy acquired from the simulation was ~33 keV. Thus, most of the applied photons were capable of exciting the K-shell electrons of silver atoms, a suitable phenomenon to consume the high energy radiation and enhance the specific attenuation contrast[6]. The X-ray spectrum was gently filtered by the polypropylene syringe used as a sample container reducing the beam hardening artifacts. As an object made of hydrocarbon, the syringe has about the same effective atomic number as the examined samples, excluding the dense silver precipitates. Thus, part of the lowest energetic photons is filtered out before reaching the ultimate FOVs, and are not incorporated into the reconstructions as superficial attenuation artifacts, usually seen with non-filtered polychromatic radiation[5,6]. However, in the superficial µCT imaging, local and moderate beam hardening artifacts were still observed; they were taken into account while adjusting the segmentation threshold used to quantify the nuclei (Supplementary Methods, Protocol 3). The imperfections in the

image capture devices and the resulting ringing artifacts were damped by using a so-called dynamic ring-removal option, which slightly changes the position of the sample between successive X-ray images[5]. During the imaging, the internal temperature of the µCT was constant at +29 C.

During the tomographic sessions, the µCT also collects so-called drift files, a chronological set of images taken from the 0° angle between one time periods. With the aid of these files and earlier experience, it was known that sample platform itself does about 5 µm back and forth swaying while rotating the full 360° circle, likely due to imperfect bearings. Thus, if larger than 10 µm sample drift was observed, the collected data were rejected, and the imaging was repeated until stable data acquisition was achieved. To ease the observation of the surface of the low-density PLA-based flat substrate immersed in the attenuating water, we aligned the sheet along the imaging pathway at 0° angle to produce sharp contrast in the preview.

The chosen exposure times and sampling density resulted in imaging times of about 24, 11, 5, and 15 h for the well-plate-based flat-substrate, dry and wet scaffold, and PLA-based flat-substrate samples, respectively. The time needed for imaging is partly the result of the high magnification and the low power of the used X-ray tube. We made no attempt to optimize the imaging time as to do so would have been out of the scope of this research. However, to enable virtually the same results, shorter imaging times could be achieved especially with higher power X-ray sources. Otherwise, we recommend interested in trying increasing acceleration voltage, decreasing the exposure time and/or the number of transmission images.

**µCT image reconstruction**. The tomographic µCT data were reconstructed into 16-bit volumes with XMReconstructor 8.1 software supplied by the system manufacturer. Center-shift correction values were chosen manually but systematically according to the principles described in Supplementary Methods, Protocol 5. To normalize and utilize the dynamic range effectively for the quantification, the highest and lowest intensity values were searched for each series by inspecting preliminary reconstructed cross-sections, an option available in the reconstruction software. The gray levels in the data series were then normalized by appropriate byte scaling intervals that easily covered the extreme values found. Byte-scaling intervals of (−1000)-(6000) and (−3000)-(50,000) and (−300)-(3000) were found to be appropriate and used to reconstruct the wet and dry imaged scaffolds, and the PLA-based flat-substrate, respectively. The kernel size of 0.7 was set for the reconstruction filter for all the samples, and all the defect correction options were disabled.

**Segmentation and pre-processing of µCT data**. The 3DROQA begins by receiving 3D grayscale data as an input starting from the reference-based adaptive segmentation (Supplementary Methods, Protocol 1). The µCT reconstructions were processed in Avizo Software (2019.3, Thermo Fisher Scientific) and MATLAB (R2019b, Mathworks), and visualizations were partially done using both software. More details about image compositions are available in Supplementary Note 4 in addition to exceptions of the software versions used during the extent of the research. The reconstructions from which the nuclei were quantified were prepared as follows.

First, an intensive ringing artifact, caused by the imperfections in the used objective (Supplementary Fig. 13) was removed by cropping 88 µm thick bottom slices from each reconstructed wet volumes affected the most. The volumes were then white top-hat transformed (WTH) and thresholded to segment the silver signals from the lower intensity voxels using the Interactive Top-Hat module. WTH is a subtraction of the opened image from the original one that can be used to extract features smaller than the given kernel size. This locally adaptive segmentation method was used since the silver signals were distributed all around the FOVs as local intensity spikes surrounded by varying background intensities. In order to adjust the WTH threshold correctly for each sample series, one sample was superficially examined with the µCT and light microscopy (Figs. 3a, 4b, c, Supplementary Fig. 10 and Supplementary Methods, Protocol 3). The seen longest and shortest axis of various superficial nuclei were manually measured from the optical images, and the corresponding voxel particles were located from the µCT reconstructions. By using the Maximum Intensity Projection module in Avizo, the µCT data were viewed from the aspect of the light microscope objective. In the segmentation editor, the WTH threshold was tuned until the seen longest and shortest axes of the optically measured nuclei and the corresponding voxel particles gave the best possible match. If the µCT particles were noisy, such as seen in Fig. 4 wet imaging without CCD pixel binning, the overall shapes were assessed. Because the silver was the densest substance in the samples, the higher WTH threshold limits were set to the maximum while the lower limits of 5400; 2700; and 13,700 were chosen for the wet and dry imaged scaffolds, and the PLA-based flat-substrate sample, respectively. In Fig. 4c, an additional data enhancement was applied using first Non-Local Means module with default settings: GPU Adaptive Manifold, 3D volume, Spatial Standard Deviation 5, Intensity Standard Deviation 0.2, Search Window 10, Local Neighborhood 3. Then segmenting using the Interactive Top-Hat module with the lower threshold limit of 5000. The kernel sizes of 5 and 10 were chosen for the wet and dry imaging, respectively, to adapt to the varying voxel size. 3D and connectivity of 26 were used as neighborhood settings for all data. Further discussion about the reference-based adaptive segmentation is given in Supplementary Methods, Protocol 3.

After the WTH segmentation, the Fill Holes module was used to ensure that no hollow voxel particles were present in the µCT data. Possible incomplete voxel particles in touch with the edges of the FOVs were excluded by running the Border Kill module. To represent all the analyzed shapes by at least a decent number of voxels and to remove small-object noise, particles smaller than 15 µm³, 100 µm³, and 13 µm³ from dry and wet scaffolds, and the PLA-based flat-substrate sample were removed, respectively. This was done using the Sieve Analysis module, after the voxel particles were analyzed with the Label Analysis module with 3D interpretation and basic measures. The number of particles in different steps was calculated by using the Label Analysis module. The number of filtered particles can be found from the Supplementary Tables 1–3. The filtered binary volumes and the cropped intensity volumes were exported from Avizo to 3D raw format and imported into MATLAB for further processing.

**Segmentation and pre-processing of CLSM data**. The CLSM data presented in Fig. 4 were processed for quantification following the principles described above. However, as the data was used as an optical 3D reference owning better resolution, we made extra efforts to ensure that only true well-resolved nuclei with good quality were present in the data. The quantified CLSM data consisted of 14 FOVs acquired from the same PLA-based flat-substrate sample. The following settings were chosen to make the segmentations fit well with the crisp nuclear fluorescence signals seen in the xy perspective, as the nuclei were not as clearly reproduced in the z perspectives (Fig. 4a). The 8-bit data were segmented using a lower WTH-threshold limit of 50 with the kernel size of 33. The normal procedure was followed with the Fill Holes and Border Kill modules. The Ball Closing was then used with the size of 3 to connect isolated nuclear voxels. The Sieve Analysis with 13 µm³ volume high-pass filter was used first at this point, to exclude the smallest, clearly non-nuclear particles from further processing. To finalize the segmentation, the Binary Smoothing with kernel size of 3 and threshold of 0.5 and Ball Closing with size 12 were used. By overall visual assessment and noting the better resolution, we believe the true well-resolved CLSM nuclei were at least satisfyingly reproduced in all dimensions for use as a reference for µCT data owning 1.1 µm/0.33 µm = 3.3 times coarser voxel dimensions. By visually investigating the segmented CLSM data, we observed a lot of small-object noise, a few bigger unknown extra-nuclear background particles (Fig. 4a), and a couple of under-segmented nuclear aggregates. Using the volumes, we looked for the lower and upper band-pass limits to exclude all the mentioned anomalies. We found a range of 120–2290 µm³ (3276–62,518 voxels) suitable for cleaning the data accurately, excluding only a few true nuclei (six in total, from which four were part of the under-segmented quasi-aggregates), leaving 82 representative particles for the statistical comparison (Fig. 4d). The final data obtained from the 14 FOVs were combined into a single data set in Excel, after the Avizo and MATLAB processing.

**Quantification of data**. In order to simplify the quantification in the 3DROQA (Supplementary Methods, Protocol 1) and minimize user burden, we developed a MATLAB code[10] that automatically executes all the particle transformations for a studied volume and tabulates the quantitative measures to be used for the statistical analyses. To transform the initial voxel particles into ellipsoids that have easily interpretable axes, the code first generates polygon particles using iso2mesh, a 3D mesh generation toolbox for MATLAB[26]. Vol2surf function was used by setting the maximum radius for the Delaunay sphere 0.5 and isovalue 1. After the mesh creation, the surfreorient function was used to arrange the meshes for further calculations. Ellipsoids were then fitted onto each individual polygon particle by using the ellipsoid fit function based on the linear least squares approach[27]. Hyperboloids were removed from the data as anomalies, recognized by their negative or complex values returned by the ellipsoid fit function.

The success of the whole particle transformation process was assessed by the ratio between the volumes of the corresponding ellipsoids and voxel particles (ellipsoid-to-voxels volume ratio). In the case of more complicated shapes than naturally smooth and roundish nuclei, the quality of the particle transformation may decline, usually seen as the high or low ellipsoid-to-voxels volume ratios. To better consider data anomalies, a large number of voxel particles were visually compared to their daughter ellipsoids (a few of them shown in Fig. 3b–d and Supplementary Fig. 11). The visual validation in mind, a 0.8—1.2 ellipsoid-to-voxels volume ratio band-pass filter was chosen to exclude low quality fits from all the data, such as particles representing under-segmented quasi-aggregates of multiple nuclei (Fig. 3g). The filter was manually applied on Excel data (Supplementary Data 2) exported by the MATLAB code[10]. The number of excluded particles from each data are represented in Supplementary Tables 1–3.

The particle analysis contained the following measurements. The nuclear volumes and mean signal intensities were calculated from the voxel particles. The measured shape properties included the radii (longest, medium, and shortest, eventually converted to diameters in R software) of the fitted ellipsoids, flatness (the ratio of the shortest ellipsoid radius to the medium one), elongation (the ratio of the medium ellipsoid radius to the longest one), and sphericity[28] (calculated by $\frac{\pi^{\frac{1}{3}}(6Volume)^{\frac{2}{3}}}{Area}$, where the Volume and Area were obtained from the polygon particles, Supplementary Notes 5, 6). In some cases, the used stlVolume function failed to calculate the volumes for the polygon particles leading to a few particles without sphericity values (the numbers are shown in Supplementary Tables 1–3). Excluding these problematic particles, we tested that the stlVolume function calculates the

same volume measurements as the surfvolume function of the iso2mesh library, an alternative function for the polygon-volume calculation. A so-called anisotropy value (one minus the ratio of the shortest radius to the longest one of the ellipsoids) was also calculated, an additional way to inspect the radii, not investigated in this paper (data available in Supplementary Data 2).

As with any quantitative method, if possible, we recommend arranging known controls to which the experimental results can be compared. For example, if the tendency of a chemical to disturb the actin cytoskeleton is studied, control samples without and with the exposure to cytochalasin D or phalloidin could be used. Referring to the literature of toxicology in this case, the control concentration series, including the known minimum concentration that is known to cause biological effect can be helpful to determine the sensitivity of the 3D image analysis. Repeating the experiments helps to assess the within-laboratory reproducibility of the experiments and the variance of the studied features.

**In silico disintegration of under-segmented aggregates**. All the previously rejected dry μCT-imaged particle anomalies (hyperboloids and particles rejected in the 0.8–1.2 ellipsoid-to-voxels volume ratio filtration as aggregates) were reanalyzed after disintegrating them in silico with a 3D marker-based watershed separation[29] method in Avizo (Supplementary Methods, Protocol 1). The MATLAB code[10] has an option to create binary volumes that can be used to point out the anomalies in Avizo. First, in order to enhance the contrast and sharpen the natural edges of nuclei, original grayscale μCT reconstructions were unsharp masked (Fig. 3f) by using the following parameters: 3D interpretation, edge size 7, edge contrast 3, and brightness threshold 0. The used Unsharp Masking module resulted in 32-bit signed image format as a default, which was scaled back into 16-bit unsigned format in the Convert Image Type module with 1 scaling and 0 offset. Next, the enhanced data were masked by the corresponding binary volumes that contained the anomalies using the Mask module.

To increase the accuracy for addressing the individual nuclei inside the under-segmented aggregates, the kernel independent H-maxima module was used instead of a second WTH segmentation to extract the local intensity spikes (Fig. 3f). In the module, 3D Interpretation was used, Neighborhood was set to 26 and contrast to 900. The following Marker-Based Watershed module, used with 3D interpretation, 26 neighborhood, watershed type, and fast mode, required topological and marker information. The previously contrast-enhanced and masked grayscale volumes, which were inverted with the NOT module, were used as flooding topology, while the H-maxima markers indicated the cores of the nuclei. After the flooding, the under-segmented aggregates were cut along the acquired watershed lines with the AND NOT Image module to extract the individual nuclei. Next, the new daughter particles smaller than 15 μm³ were excluded by the Sieve Analysis module as carried out previously. The reprocessed volumes were converted to the 3D raw format and imported back for the developed MATLAB code[10] for reanalysis. The data filtration flow in the reanalysis is represented in the Supplementary Table 2.

**Statistics and reproducibility**. The tabulated quantitative data, exported as XLS files by the developed MATLAB code[10], was converted to CSV files by Excel (1908, Microsoft) after data management (acquiring intermediate figures and applying the data filtrations such as 0.8–1.2 ellipsoid-to-voxels volume ratio). The CSV files were opened by R software (3.6.2, The R Foundation) in which the statistical significances and box-plots were analyzed by using a script file (Supplementary Data 1).

The data distributions were assumed to be skewed since the values were confined by the analytical limits (the volume high-pass filters and axial ratios limited to interval ([0, 1])). Furthermore, the data obtained from the scaffold samples contained multiple far-reaching outliers passed through the systematic data filtration (Figs. 3g, 4d, Supplementary Figs. 8, 9). Thus, instead of further cleaning the data and using statistical tools based on averages, we used medians, SIQR, and two-sided Wilcoxon-Mann-Whitney test to assess the data. The large number of the particles acquired from the scaffold samples (Supplementary Tables 1–3) usually resulted in very small $p$ values (Fig. 3g, Supplementary Figs. 8, 9) below the conventional significance limit of 0.05. Thus, the obtained $p$ values were also compared to the general level of the $p$ values in the experiments. Moreover, because the $p$ values can be very small while the sample populations clearly overlap, we wanted to further evaluate the distinctiveness between the compared distributions. We considered the distributions to be well resolved if the median difference of a considered value was larger than the average SIQR calculated for the compared distributions. In the case of the axis we considered it convenient to further proportion the median differences and SIQRs to the used voxel size. Most of the detailed statistical discussion based on these principles can be found from the Supplementary Note 5.

**Reporting summary**. Further information on research design is available in the Nature Research Reporting Summary linked to this article.

## Data availability

To explore the principles of the 3DROQA and to make a test run, a subvolume of a dry-imaged control scaffold is provided with the MATLAB code[10]. The whole 3D-imaging data are available from the corresponding author upon request via email. The μCT data are stored as RAW files, the optical data as TIF sets. We would like to note that the data sets are somewhat large. As an example, the size of the μCT-imaged wet-state scaffold-sample set is 21 GB. For the dry-scaffold data set, the size is 167 GB.

## Code availability

The MATLAB code that was used to run the automatic operations of the 3DROQA (Supplementary Methods, Protocol 1) is available via Zenodo repository[10].

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

## Acknowledgements

We would like to thank Sari Kalliokoski and Hanna Vuorenpää for the consultation, management, and delivery of cell samples. Mart Kroon for his help with the PLA-based flat-substrate samples. Toni Montonen and Olli Koskela for their help in designing and performing the OPT imaging and reconstruction. Nathaniel Narragirish for his invaluable help in the digital data analysis. Jari Viik for his statistical consultation. Finally, we would like to thank Business Finland and the Council of the Tampere region for financing the Human Spare Parts project, Academy of Finland (Decision numbers: 308315 and 314106), and Academy of Finland Center of Excellence on Body on Chip Research (Decision number: 312412).

## Author contributions

I.T. performed antibody labeling and other related laboratory work with M.O. M.O. performed part of the hASC-culture-related work. The responsible person for the Caco-2 culture and the design of the fibronectin coating was T.I. Optical microscopy examinations were made by I.T. and M.O. I.T. optimized and performed μCT imaging and reconstructions. I.T. set the detailed analytical goals and contributed to the design of the digital analysis processes planned, made, and executed by M.H. and K.L., who also acquired most of the μCT visualizations. I.T., K.L., and M.H. handled the μCT data and are the responsible persons for data integrity. M.H. designed, wrote, and tested the MATLAB code. K.L. designed and performed the disintegration of the under-segmented aggregates according to the general guidance and technical help given by I.T. and M.H., respectively. I.T. planned the statistical analyses together with A.A., and interpreted the results and made the conclusions helped by M.H., K.L., A.A., and J.H. The literature research was made by I.T. who also carried out the major writing work, designed and paginated the figures, planned the main structure and experiments of the article and coordinated the research project on a practical level according to the guidance given by J.H., A.A., and T.I. The idea of using antibody-silver labeling to enhance subcellular features in μCT-imaged biomedical samples was proposed by J.H., who initiated the research aiming to extend the analytical capacity of tube-source μCT. The idea of using cytochalasin-D treatment to demonstrate the 3D quantification of a subcellular phenomenon was proposed by T.I. Cell-biological resources and advice were given by S.M., T.I., and M.O. Biomaterial resources and advice were given by M.K. and L.J. T.I. and A.A. provided consultation for the optical and μCT imaging, respectively.

## Competing interests

The authors declare no competing interests.
