## [Peer Review File · Communications Biology]

Reviewers' comments:

Reviewer #1 (Remarks to the Author):

The main body of the work utilizes "optically difficult samples", including bioengineering scaffolds and organs-on-a-chip, to compare the relative strengths of various 3D imaging methods. A significant body of work is contained in the main body of the paper, but

This manuscript is an attempt to achieve the challenging and laudible goal of establishing the potential power of X-ray micro tomography (micro-CT) imaging for assessing 3-dimensional cellular features as compared with other imaging modes that are based on visible light. The main body of the work utilizes "optically difficult samples", including bioengineering scaffolds and organs-on-a-chip, to compare the relative strengths of the alternative 3D imaging methods. The authors do an excellent job in the supplemental notes with describing the foundational principles, advantages and disadvantages of different imaging modes. A comparison of microCT with traditional imaging methods is provided to address the effectiveness of subcellular component staining and advantages of increased penetrance that microCT provides. As the community works towards using commercially-available microCT instrumentations for the study of cell characteristics in "deep tissue" that lie beyond the reach of light-based imaging methods. The paper has strengths but is in need of figure and experimental improvements, as well as a quality of writing with the level of clarity, precision, and specificity that would be suited to this journal.

Major issues:

Characterizing cellular features and change in complex tissues must, as the authors point out, avoid disrupting the complex organization of such systems. To do this, the authors use the antibody-mediated silver deposition to label large cellular features such as nuclei or cytoskeletal elements and show an example of assessing one of these systems in terms of nuclear size/shape with or without cytochalasin-D treatment. This work is in need of positive and negative controls. Samples would optimally be compared between the subject method and a gold-standard method such as reconstruction from electron microscope-based block face or serial section, or potentially, super-resolution light sheet imaging. After the error for the controls is established, the differences in 3D measurements across thousands of cells from a single micro-CT scan would be more meaningful. Dependence of modeling on resolution required to image the specific structure would best be noted specifically.

The first figure demonstrates that silver deposition using the EnzMet system produces similar patterns in 2D cell culture as similar elements labeled with fluorescent markers. This is a good comparison to make, but some relatively simple optimizations could have been tested to diminish what the authors call "streaking artifact", which is more traditionally called "beam hardening artifact". This problem could potentially be addressed by higher energy x-rays or by lowering the maximum staining. We note that the "silver reactions were not observed with a light microscope... Instead, the recommended maximum silver generation time was used to maximize the signal intensities." The authors also comment on the "grainy" appearance of the generated silver label (vs. phalloidin labeling). It is not clear if these are inherent properties of the silver deposition, the antibody distribution, or non-specific effects of staining time. They note that some cellular features which were theoretically resolvable were masked by artifacts. To what extent were these issues due to the limits of resolution, or a need for some basic optimization of staining or use of synchrotron sources?

Figure 1. While the first figure included comparisons with fluorescent imaging, the second figure (regarding 3D cell culture environments) includes samples which have only been stained with the antibody-silver method and then assessed with OPT, confocal laser scanning microscopy, and microCT. They note that only microCT can penetrate the entire sample with sufficient z-resolution. Nonetheless, it would highly useful to directly compare images derived from silver-labeled microCT to fluorescently-labeled samples imaged with confocal and light-sheet microscopy. Essentially, one is labeling different cell components using the different methods, but a nucleus should theoretically look like a nucleus regardless of stain and imaging mode. To facilitate proper comparison between methods, it would be better to prepare a supplementary figure in which structures are uniformly represented – for example, all nuclei and membranes dark, and background white.

Other issues:

Title: Not clear from the title what the authors mean by probative sampling

Abstract "cell structure-specific labeling" is a bit too vague.

"Carefully" is overused. It is better to just show how the work was careful.

By "special reach" do the authors mean field of view? How does that differ from "coverage"?

The words "delicate biomechanical phenomenon" is vague.

Why use "extremely" or "large" when referring to statistical significance"?

The last sentence first paragraph is miswritten.

Introduction: "Advanced"? in what way?

Figure 2 – 3D views of the data may better showcase the power of micro-CT imaging. It would be interesting to see how the edges of cell nuclei compare with each of the methods in a close-up view

Figure 3 – Ideally, at least a subset of the segmented nuclei should be verified by manual counting. It is also puzzling how are the p-values I can be in the range of 10^{-16} when the boxplots overlap the control and experimental groups.

100: Proper usage would be "invert" rather than "inverse"

110-111: What was the basis for choice of the upper and lower limits in histogram levels?

145: Please specify "reasonable"

168: It would be more accurate to use "no binning" rather than "1 x 1 CCD pixel binning"

178: Rounded object detection was based on what principles? Were there positive and negative controls?

S10 legend – the figure panels are not labeled

When discussing increasing performance in supplementary note 3, it would be beneficial to elaborate on what exactly (resolution, signal strength, scan time, etc.) is being improved.

Additionally, in addition to showing "native" imaging formats, it would be useful for the sake of direct comparison, for the figure 1 images to be alternatively presented in a way that same structures are represented the same way (black or white).

The paper is predicated on the idea of demonstrating "the undiscovered potential of a commercial X-ray tube source [micro]CT" but this wording implies novelty in the use of non-synchrotron microCT. The breadth of prior microCT work can be better acknowledged. However, this cross-modal comparison is novel.

Reviewer #2 (Remarks to the Author):

This manuscript represents a convincing account of a tube-based 3D and micron resolved X-ray absorption imaging modality for scaffold supported cellular organelles in turbid environments resting on silver binding to antibodies, in turn binding to specific subcellular sites, and involving a meticulous and well thought-out sample preparation and data processing pipeline.

While the methodological aspects are well explained (there should be a bit of an English overhaul in the main text) I am missing some more definitely exposed venues for applications of the method, and the authors should try to motivate the reader better in this respect. Also, it would be fair to mention explicitly the potentials of structured illumination, light-based fluorescence microscopy (e.g. Reynault et al, Nature Methods 12, 30-34 (2015) and refs. therein) and propagation based X-ray phase contrast (e.g. Moosmann et al, Nature (2013)) which both achieve a spatial resolution of the order of the present method but require hardly any sample preparation (apart from fluorophore expression in the light-sheet case), introduce much less radiation toxicity and achieve temporal resolutions in the sub-minute range for 4D in vivo imaging.

I do believe that the present method has a clear potential for biological and medical research, partly due to its wide-spread applicability, partly due to the highlighting of specific membranes, and mostly due to the deeply penetrating volume capability and its potential for sub-micron spatial resolution if X-ray beam divergences are exploited with very small focal spot sizes to control the associated blur. The authors should also mention a third alternative for X-ray sources that could be

readily available in the foreseeable future: plasma-wakefield accelerators for electrons in combination with magnetic structures to release the radiation.

Therefore, after a mild revision doing justice to my above points, I recommend this ms for publication in Nature Communication Biology.

Response letter to peer review – Submission COMMSBIO-19-0548-T

On behalf of all the authors involved in this study, I would like to express our gratitude for the time the editor and reviewers have put into the valuable comments concerning our manuscript. We thank the reviewers for their excellent suggestions and constructive criticism, which helped us to make the manuscript better. We have done our best to respond to the revision requests with our best abilities, and we hope the editor and the reviewers are pleased with our revised manuscript.

All the detailed comments to revise the manuscript and our responses to them are listed below. Also, as recommended by the reviewers, we improved the overall quality and clarity of the text and message delivery in general. After the responses, we provide a list of details about other revisions we did for the manuscript.

Reviewer comments and responses:

Reviewer #1

The main body of the work utilizes “optically difficult samples”, including bioengineering scaffolds and organs-on-a-chip, to compare the relative strengths of various 3D imaging methods. A significant body of work is contained in the main body of the paper, but

This manuscript is an attempt to achieve the challenging and laudible goal of establishing the potential power of X-ray micro tomography (micro-CT) imaging for assessing 3-dimensional cellular features as compared with other imaging modes that are based on visible light. The main body of the work utilizes “optically difficult samples”, including bioengineering scaffolds and organs-on-a-chip, to compare the relative strengths of the alternative 3D imaging methods. The authors do an excellent job in the supplemental notes with describing the foundational principles, advantages and disadvantages of different imaging modes. A comparison of microCT with traditional imaging methods is provided to address the effectiveness of subcellular component staining and advantages of increased penetrance that microCT provides. As the community works towards using commercially-available microCT instrumentations for the study of cell characteristics in “deep tissue” that lie beyond the reach of light-based imaging methods.

Comment 1

The paper has strengths but is in need of figure and experimental improvements, as well as a quality of writing with the level of clarity, precision, and specificity that would be suited to this journal.

Response 1

We did our best to improve the overall quality and clarity of the text and message delivery in general.

Comment 2

Major issues:

This work is in need of positive and negative controls. Samples would optimally be compared between the subject method and a gold-standard method such as reconstruction from electron microscope-based block face or serial section, or potentially, super-resolution light sheet imaging. After the error for the controls is established, the differences in 3D measurements across thousands of cells from a single micro-CT scan would be more meaningful. Dependence of modeling on resolution required to image the specific structure would best be noted specifically.

Response 2

We thank the reviewer for this constructive suggestion that helped us to improve our methodological approach and its reporting a lot. Similar to the qualitative comparison of the cell structures imaged already with different microscopy techniques in Figure 1, we now quantitatively assessed a new set of hASC cells cultured on flat biomaterial substrates (new Figure 4). These optically easy-to-image subjects enabled us to quantitatively better assess the accuracy, potential, and on the other hand, limitations of our μ CT method in comparison to an accurate 3D control data acquired by a confocal laser-scanning microscope (CLSM). We found similarities and dissimilarities from the data, and we were able to enhance our approach further focusing on data processing, and define the performance of our method in detail by taking better into account the inherent image properties of the micro-focus X-ray tube-source μ CT imaging. Now the promising method has even better and more accurately defined technical foundations, providing helpful additional information for the potential researchers utilizing our method in the future. For more detailed information about the findings and conclusions, please refer to the new Figure 4 under the section “Quantitative comparison of nuclei on flat substrate”, and corresponding parts of the “Discussion” section.

Comment 3

The first figure demonstrates that silver deposition using the EnzMet system produces similar patterns in 2D cell culture as similar elements labeled with fluorescent markers. This is a good comparison to make, but some relatively simple optimizations could have been tested to diminish what the authors call “streaking artifact”, which is more traditionally called “beam hardening artifact”. This problem could potentially be addressed by higher energy x-rays or by lowering the maximum staining. We note that the “silver reactions were not observed with a light microscope... Instead, the recommended maximum silver generation time was used to maximize the signal intensities.” The authors also comment on the “grainy” appearance of the generated silver label (vs. phalloidin labeling). It is not clear if these are inherent properties of the silver deposition, the antibody distribution, or non-specific effects of staining time. They note that some cellular features which were theoretically resolvable were masked by artifacts. To what extent were these issues due to the limits of resolution, or a need for some basic optimization of staining or use of synchrotron sources?

Response 3

To investigate possible labeling issues, an additional silver-generation experiment with anti-lamin labeled hASC on a flat-substrate sample was video recorded with silver-safe red-light microscopy (Methods and Supplementary Video 1). In prior *in vitro* test, it was verified that the red light did not produce visible Tyndall-effect, an indication about colloidal silver observed with a laser pointer, nor visible black precipitates, in the silver-generation reaction mixture during a relevant time period. The result is similar to the corresponding control solution kept in the dark, excluding intermittent observation of the Tyndall effect and possible precipitates. In the video, it can be seen that the silver

signal fits well with the expected nuclear circumference of a faintly visible subject nucleus before the start of the experiment. It can also be seen that the silver generation time even beyond the recommended maximum of 15 min did not cause any visible background development (longest time point in the video being 18 min). Furthermore, the streaking artifacts seem to be related to dry samples. In the wet-state-imaged samples, no streaking was observed, as shown in Supplementary Figure 10 and new Figure 4b and c. It is also notable how in both Figure 4 CLSM and μ CT images the nuclear signals vary likely due to varying lamin expressions. This observation further encourages to develop the silver signals well so that as many nuclei could be captured in the data as possible, even though some of the cells manifest weaker lamin expression levels. If the lamin-expression levels themselves would be under the investigation, which is not the case in the manuscript, then the maximum-achievable-signal approach would not be usable. In addition to the streaking artifacts, we believe the penumbra, which average size in Figure 1 is 0.9 μ m as pointed out in the Methods section, is another reason why the smallest curly CaCo-2-boundary features were not reproduced in the μ CT data. To image relatively large samples, one of the strengths of the tube-source μ CT devices, the geometric formation and magnification of the penumbra cannot be avoided even though the labeling itself would be ideal. To capture a large number of cells in relatively large volumes with microscopic accuracy requires balancing between the benefits and disadvantages, an unavoidable step typical when dealing with any technology.

Comment 4

Figure 1. While the first figure included comparisons with fluorescent imaging, the second figure (regarding 3D cell culture environments) includes samples which have only been stained with the antibody-silver method and then assessed with OPT, confocal laser scanning microscopy, and microCT. They note that only microCT can penetrate the entire sample with sufficient z-resolution. Nonetheless, it would highly useful to directly compare images derived from silver-labeled microCT to fluorescently-labeled samples imaged with confocal and light-sheet microscopy. Essentially, one is labeling different cell components using the different methods, but a nucleus should theoretically look like a nucleus regardless of stain and imaging mode.

Response 4

Thank you for pointing out this important aspect. As described in the Response comment 2, we made a new detailed quantitative comparison of μ CT and CLSM imaged nuclei labeled with silver and fluorescent antibodies, respectively. The new data is represented in the new Figure 4 and the results discussed in the corresponding sections of the text.

Comment 5

To facilitate proper comparison between methods, it would be better to prepare a supplementary figure in which structures are uniformly represented – for example, all nuclei and membranes dark, and background white.

Response 5

To facilitate the proper comparison between the methods, part of the Figure 1 images were directly inverted to match the gray scales. Similarly, the new Figure 4 images are presented in the same way,

expect the 2D cross-sectional images which we felt to be visually more functional with the non-inverted intensities shown with the segmentation interfaces (similar approach as used in Fig. 3a).

Comment 6

Other issues:

Title: Not clear from the title what the authors mean by probative sampling

Response 6

Together with the other general revisions to clarify the text and improve message delivery, the title was changed, and the word “probative” was not used.

Comment 7

Abstract” “cell structure-specific labeling” is a bit too vague.

Response 7

Similar to Response comment 6, the expression was omitted.

Comment 8

“Carefully” is overused. It is better to just show how the work was careful.

Response 8

We would like to again thank for these valuable tips that indeed helped us to make the reporting more accurate and efficient. Thus, in the new revised text, we avoided the use of “carefully” and paid attention to a clear description of what was actually done.

Comment 9

By “special reach” do the authors mean field of view? How does that differ from “coverage”?

Response 9

“Special reach” was replaced by more definite expressions.

Comment 10

The words “delicate biomechanical phenomenon” is vague.

Response 10

The use of “Delicate biomechanical phenomenon” was omitted.

Comment 11

Why use “extremely” or “large” when referring to statistical significance”?

Response 11

The use of expressions “extremely” and “large” was avoided when discussing statistical significance.

Comment 12

The last sentence first paragraph is miswritten.

Response 12

The miswriting was corrected.

Comment 13

Introduction: “Advanced”? in what way?

Response 13

The meaning of the expression “advanced” is explained at the beginning of the paragraph in question.

Comment 14

Figure 2 – 3D views of the data may better showcase the power of micro-CT imaging. It would be interesting to see how the edges of cell nuclei compare with each of the methods in a close-up view

Response 14

To enable a detailed comparable view between the imaging methods, in the new Figure 4a,b and c close-up views of the μ CT and CLSM imaged nuclei labeled with the same primary antibodies are shown.

Comment 15

Figure 3 – Ideally, at least a subset of the segmented nuclei should be verified by manual counting.

Response 15

In the Supplementary Protocol 3, we randomly chose 13 optically viewable nuclei from various depths of the dry-imaged PLA-scaffold. All the chosen nuclei were also found from the μ CT data. As noted in Supplementary Protocol 3, by visual comparison with the optical reference images, we considered that 9/13 nuclei segmented from the μ CT data manifested near-perfect fit, and the rest of the nuclei satisfyingly represented the overall dimensions and geometry of their optical references. In connection with the Figure 3 b,c,d,e and Supplementary Figures 5, 6, 7, 11, 12 we discuss how well the voxel shapes turned into the ellipsoids during the 3DROQA processing.

Comment 16

It is also puzzling how are the p-values I can be in the range of 10^{-16} when the boxplots overlap the control and experimental groups.

Response 16

Large samples can produce statistically significant results, even from small differences between the compared distributions. The p-value gives an estimation of how likely the observed difference is caused by coincidental bias during the sampling. Thus, p-value is not a good indicator of how much the distributions are resolved, although the p-values from similar comparisons can be compared to some extent. That is why we considered the relative magnitudes of the p-values, and further assessed how well the compared control and experimental distributions were resolved, usually assessing the median differences.

Comment 17

100: Proper usage would be “invert” rather than “inverse”

Response 17

The expression was corrected.

Comment 18

110-111: What was the basis for choice of the upper and lower limits in histogram levels?

Response 18

All the image adjustments aimed to enable a comprehensive and clear presentation of the data. For example, inappropriate adjustment of the histograms would have resulted in filling the view due to enhanced background voxels, disappear of all the investigated features, or bad contrast of the investigated features. The chosen image adjustments were found to result in the opposite, good overall representations of the data.

Comment 19

145: Please specify “reasonable”

Response 19

The expression “reasonable” was better specified.

Comment 20

168: It would be more accurate to use “no binning” rather than “1 x 1 CCD pixel binning”

Response 20

The issue was corrected.

Comment 21

178: Rounded object detection was based on what principles? Were there positive and negative controls?

Response 21

The principles and algorithms used to transform the voxel particles into the corresponding ellipsoids is described in detail in the Methods section, and further discussed in the Supplementary Note 5. We considered that the best way to verify the proper overall transformation process was to randomly sample and manually observe a few cases, and to assess the overall success of the rest of the thousands of transformations use the so called ellipsoid-to-voxel volume ratios. Example transformation cases are presented in the Figure 3b,c,d (dry μ CT imaging), and examples of direct 3D superimpositions are shown in the Supplementary Figure 11 (wet μ CT imaging). As the examples propose, most of the ellipsoids fit well onto the original voxel particles, but some deviations were assumed from the automatic process. To assess the overall success of the transformation process applied on thousands of particles and to exclude obvious misfits, graphs were drawn (Supplementary Figures 5, 7 and 12) and filters used on the ellipsoid-to-voxel volume ratios, respectively. As the graphs propose, most of the ellipsoid-to-voxel volume ratios were near one indicating that most of the transformations were good, where the ellipsoids inherited the overall similar volumes and geometries of the original voxel particles. The minor cases outside the range 0.8-1.2 ellipsoid-to-voxel volume ratios were excluded, the number of rejected particles accurately reported in the Supplementary Tables 1, 2 and 3.

Comment 22

S10 legend – the figure panels are not labeled

Response 22

The figure panels were now labeled.

Comment 23

When discussing increasing performance in supplementary note 3, it would be beneficial to elaborate on what exactly (resolution, signal strength, scan time, etc.) is being improved.

Response 23

In addition to the mentions about the enhanced properties, such as increased resolution, better contrast, and shorter exposure times, in supplementary note 3 we also describe example imaging subjects and how accurately they were studied with the more sophisticated sources.

Comment 24

Additionally, in addition to showing “native” imaging formats, it would be useful for the sake of direct comparison, for the figure 1 images to be alternatively presented in a way that same structures are represented the same way (black or white).

Response 24

As stated in the response comment 5 the structures are now presented the same way.

Comment 25

The paper is predicated on the idea of demonstrating “the undiscovered potential of a commercial X-ray tube source [micro]CT” but this wording implies novelty in the use of non-synchrotron microCT. The breadth of prior microCT work can be better acknowledged. However, this cross-modal comparison is novel.

Response 25

On the Supplementary Note 3, the general level of tube-source μ CT usage in scientific research is described. In the newly revised material, we took even further the cross-modal comparison (new Figure 4) to demonstrate the benefits of the above-general-level usage of tube-source μ CT.

Reviewer #2

This manuscript represents a convincing account of a tube-based 3D and micron resolved X-ray absorption imaging modality for scaffold supported cellular organelles in turbid environments resting on silver binding to antibodies, in turn binding to specific subcellular sites, and involving a meticulous and well thought-out sample preparation and data processing pipeline.

Comment 1

While the methodological aspects are well explained (there should be a bit of an English overhaul in the main text) I am missing some more definitely exposed venues for applications of the method, and the authors should try to motivate the reader better in this respect.

Respond 1

We would like to thank the reviewer for the encouraging words and constructive views, which helped us to make the manuscript better. We did our best to refine the overall quality of the manuscript, clarify the text, and enhance message delivery. To better motivate the reader, immediately at the abstract, we listed common material properties that could cause imaging issues, problems that could be solved using the X-ray approach presented in the manuscript.

Comment 2

Also, it would be fair to mention explicitly the potentials of structured illumination, light-based fluorescence microscopy (e.g. Reynault et al, Nature Methods 12, 30-34 (2015) and refs. therein) and propagation based X-ray phase contrast (e.g. Moosmann et al, Nature (2013)) which both achieve a spatial resolution of the order of the present method but require hardly any sample preparation

(apart from fluorophore expression in the light-sheet case), introduce much less radiation toxicity and achieve temporal resolutions in the sub-minute range for 4D in vivo imaging.

Respond 2

In addition to light-sheet techniques, we took into account structured-illumination microscopy in Supplementary Note 1. In Supplementary Note 2 and 3, we added the reference to Moosmann et al, Nature (2013) adjacent to the discussion about phase contrasting and synchrotron sources, respectively.

Comment 3

I do believe that the present method has a clear potential for biological and medical research, partly due to its wide-spread applicability, partly due to the highlighting of specific membranes, and mostly due to the deeply penetrating volume capability and its potential for sub-micron spatial resolution if X-ray beam divergences are exploited with very small focal spot sizes to control the associated blur. The authors should also mention a third alternative for X-ray sources that could be readily available in the foreseeable future: plasma-wakefield accelerators for electrons in combination with magnetic structures to release the radiation.

Therefore, after a mild revision doing justice to my above points, I recommend this ms for publication in Nature Communication Biology.

Response 3

In Supplementary Note 3, instead of only referring to “exotic apparatuses”, we added more definite expressions that we referred to both plasma-wakefield accelerators and regenerative debris-free droplet-target laser-plasma X-ray sources.

Reviewer #3

Remarks to the Author:

This manuscript describes the development of an x-ray imaging and analysis pipeline for visualizing and quantifying sub-cellular structures in specimens that are difficult to image using existing modalities. The described methods fill an information gap by imaging specimens, such as tissues, that are too voluminous for light microscopy (i.e. larger than 2mm). The authors applied their techniques to imaging large numbers of stem-cell nuclei in tissue-engineering scaffolds. The described methods have the potential for wide applicability. This manuscript is therefore of interest to a wide readership, and of profound interest in a number of medical and research areas, such as screening candidate pharmaceuticals, tissue engineering, development of organ-on-a-chip platforms, and so on. Moreover, there is overwhelming evidence that cultured cells deviate in sub-cellular organization from the same cell type in the context of tissue.

Given the potential importance of this manuscript, I recommend it be published provided the following concerns are addressed satisfactorily.

Comment 1

1. High resolution x-ray and electron microscopy studies of individual cells have shown chemical fixation with aldehydes perturbs – often radically perturbs – the sub-cellular structure. While the morphology of cells imaged by x-ray resembled DAPI stained cells imaged by fluorescence, both sets of specimens had been chemically fixed (with relatively high concentrations of formaldehyde). So, while demonstrating self-consistency of data, this does not indicate the specimens imaged represent the actual in vivo state. At a minimum, the authors should address the potential change in cell architecture in the text.

Response 1

We would like to thank the reviewer for the very helpful comments that helped us to clarify our manuscript and make it better. By the aid of the comments, we noticed and fixed the unclear description of the cell fixations. We decided to dedicate a new sub-section (“Cell fixation”) in the Methods to describe the cell fixations more definitely. As explained in the new sub-section, conventional, widely accepted fixative of 40mg/ml formaldehyde (in other words 4% w/v formaldehyde, same as 10% formalin) in phosphate-buffered saline was used utilizing two sources of formaldehyde. One source demonstrates a more conventional self-preparation approach, one demonstrates a more convenient commercial ready-made solution. A partial reason for the use of slightly different fixatives is the long time span of this project due to workload and different within-institute laboratory routines. Despite the different variations of the principally same fixative, the samples in an experiment are strictly comparable as described in Methods, because the same fixative was used within an experiment.

Comment 2

2. The progression of the silver enhancement step was not monitored by light microscopy. Instead, “the recommended maximum silver generation time was used to maximize the signal intensities”. While this approach is acceptable in proof-of-concept experiments, where the absolute degree of staining is not important, the authors should address the fact that simply using the maximum silver generation time could lead to serious over-staining, and therefore inaccurate measurements of the labeled object, in this case nuclear size and shape.

Positive comments:

The ‘in silico’ methods appear well thought out and appropriate for analyzing microCT data from tissue specimens.

This is a nicely clear, well written manuscript. General readers will find the supplementary notes enormously helpful. For example, the ‘introductions’ presented in supplementary notes 1-3 will be welcome background and context for non-specialists seeking to understand this paper.

Despite my reservations that chemical fixing and over-enhancement of silver may have perturbed the native state, I find this paper a potentially compelling use of a commonly available x-ray instrument. In the future, the developed methods may have great potential in many spheres.

Response 2

We thank the reviewer for the positive remarks! To assess possible issues relating on the development of the silver signal, a new silver-generation experiment on antibody labeled hASC was arranged and video recorded with a light microscope. From the Supplementary Video 1 (see also the related Method section) it can be observed that the developed silver signal fits well with the

expected nuclear boundaries faintly visible before the experiment, and produce no background signals even after exceeding the generation time used in the paper.

Additional details

In addition to requested corrections and given suggestions, we did additional checking of possible errors while revising the manuscript. Below are listed a few more specific revisions we made.

Detail 1

We keep the following observation insignificant but wanted to openly report the issue we found and change the expressions we used accordingly. The earlier expression “to represent all the studied shapes by at least 10 voxels” was slightly incorrect for the dry scaffold μ CT data. The accurate voxel size in figure 3, for example, was $1.10114 \mu\text{m}$, corresponding to the accurate volume of $(1.10114 \times 1.10114 \times 1.10114) \mu\text{m}^3 = 1.33514249016 \mu\text{m}^3$. Using the $15 \mu\text{m}^3$ high-pass limit, the smallest non-rejected particles were actually 12 voxels large. The $100 \mu\text{m}^3$ high-pass limit used for the wet scaffold data was correct in light of the expression. The accurate voxel volume was $(2.20227 \times 2.20227 \times 2.20227) \mu\text{m}^3 = 10.6809944208 \mu\text{m}^3$, meaning that the $100 \mu\text{m}^3$ high-pass limit will truly pass 10 voxels large or larger objects further. The found error does not fundamentally change the usability of the $15 \mu\text{m}^3$ high-pass filter. We considered that at least 10 voxels are needed to represent the studied shapes, and the smallest number was actually 12 voxels. Furthermore, the volume change caused by the two-voxel shift is only $1.33514249016 \mu\text{m}^3 \times 2 / 347 \mu\text{m}^3 \times 100 \% = 0.77 \%$ from the control median volume of the dry scaffold (the distribution is shown in Figure 3g). For the dry particles extracted from the under-segmented quasi-aggregates the percentage is $1.33514249016 \mu\text{m}^3 \times 2 / 203 \mu\text{m}^3 \times 100 \% = 1.32 \%$ (the distribution is shown in Supplementary Figure 8). The previous expression was replaced by a new one: “to represent all the studied shapes using at least a decent number of voxels”. We hope this expression is appropriate.

Detail 2

We noticed playback problems with the previously submitted supplementary videos. That is why QuickTime Player 10.5 (935.5) was used to export the earlier mpg-based videos into mov files with 1080p quality. This procedure also helped to reduce the file sizes remarkably for easier distribution online. The quality decreased a little, but not to the extent interfering the assessment of the visualized data.

Detail 3

The following clause was removed from the methods “Due to limitations in the used R software and very significant differences in some of the compared distributions, the same smallest possible p-value of $< 2.2 \times 10^{-16}$ was reproduced for multiple statistical tests.”. This was done because we observed that R can actually output even more accurate p-values. However, as the p-values go this far below the conventional significance limit, we did not consider more accurate values to be relevant for assessing the data.

Detail 4

The title was modified according to the format guidelines. We also paid attention on the clarity of the message, and how to better deliver the value of our research.

Detail 5

The abstract was modified according to the format guidelines, in addition to the clarification and enhancement of the value delivery.

Detail 6

The main text of the paper was rearranged and formatted to meet the formatting guideline. The provided section order was followed: Title, Abstract, Introduction, Results, Discussion, Methods, Code availability, References, Competing interest statement. Author information, Author contributions statement, and Acknowledgements are provided in the cover letter.

Detail 7

Appropriate parts of the manuscript were parsed and modified as necessary under the Results section. Suitable sub-headings were used to enhance the overall readability of the work. The material of the previous manuscript was also revised keeping the new Figure 4 related experiment on mind, referring to it if necessary and making the overall manuscript coherent with the determined flow.

REVIEWERS' COMMENTS:

Reviewer #1 (Remarks to the Author):

The authors seek to show the utility of one common microCT machine from Zeiss, comparing analytical power for imaging large stem cells in comparison with light microscopy. We know that light has high resolution, but poor depth penetration, while microCT has had lower resolution but greater depth penetration. The worthwhile goal is to show the utility of microCT, with supportive evidence from light microscopy. In addition the paper is accompanied by useful summaries of the various 3D imaging technologies.

An important question raised by this paper is whether the presented work has significant potential relevance to scientists studying smaller cell types or tissue samples. Studying cells grown on solid substrates is so much a specialty application that demonstration of utility beyond their setting would be indicated.

Given the relatively large voxel sizes and large sample-size limitations, the resulting images were unconvincing, particularly with the strikingly large size bars of 100 μm . Showing results for relatively large cells begged for a discussion applicability to smaller cell types. This important limitation suggests marginal utility for other biological scientists, which leads to the conclusion that this paper would seem more appropriate for a specialty journal. In discussions of resolution, it would have been useful to indicate how the sizes of their cells compare with the sizes of other cell types. It would also have been useful to indicate the dimensions of specific biological structures whose sizes determine the resolution needed to see them. A voxel size of 1.2 μm , for example, would theoretically allow detection of objects of 2.4 μm in size. Therefore, the claim of 1.2 μm voxel resolution needs to be supported by showing resolution 2.4 μm cell or man-made features.

The writing can also be more clear. The important concluding sentence of the abstract was particularly puzzling:

"undiscovered potential of the readily available x-ray tomography devices that can be used to study relatively large cell-cultured applications thoroughly with 3D microscopic accuracy."

What is "relatively large cell-cultured applications thoroughly"?

In sum, the general importance of the work does not seem to be of sufficiently large impact scientifically or methodologically to warrant publication in *Communications Biology*.

Reviewer #2 (Remarks to the Author):

The authors have undertaken a great effort to improve their ms. They have addressed the points raised in my previous report, and as far as I can see, also those by the other two reviewers in a satisfactory way. The ms meets the high standards of the journal now and should be published. I recommend the present ms for publication in *Nature Communications Biology*.

side remark: In the first sentence of the Summary section I read "with visually and quantitatively" which should read "visually and quantitatively". Also, this statement occurs more or less twice in the summary section, please change this.

Reviewer #3 (Remarks to the Author):

The authors have addressed my concerns. In my opinion, the manuscript is now suitable for publication in Communications Biology.

Reviewer comments and responses:

Reviewer #1

The authors seek to show the utility of one common microCT machine from Zeiss, comparing analytical power for imaging large stem cells in comparison with light microscopy. We know that light has high resolution, but poor depth penetration, while microCT has had lower resolution but greater depth penetration. The worthwhile goal is to show the utility of microCT, with supportive evidence from light microscopy. In addition the paper is accompanied by useful summaries of the various 3D imaging technologies.

Comment 1

An important question raised by this paper is whether the presented work has significant potential relevance to scientists studying smaller cell types or tissue samples. Studying cells grown on solid substrates is so much a specialty application that demonstration of utility beyond their setting would be indicated.

Given the relatively large voxel sizes and large sample-size limitations, the resulting images were unconvincing, particularly with the strikingly large size bars of 100 μm . Showing results for relatively large cells begged for a discussion applicability to smaller cell types. This important limitation suggests marginal utility for other biological scientists, which leads to the conclusion that this paper would seem more appropriate for a specialty journal. In discussions of resolution, it would have been useful to indicate how the sizes of their cells compare with the sizes of other cell types.

Response 1

We want to thank the reviewer for the important constructive comments. To help the readers to assess the capability of the μCT imaging concerning the varying sizes of different cellular subjects, in the Discussion, we now refer to the review article¹¹ that focuses on the sizes of different cell types. We point to figure 1 of the review article, where five different cell types are conveniently drawn for size comparison. The illustrated cells in figure 1 are claimed to be drawn to scale, and the scaling seems to be correct considering true histological images published and taking into account cellular variability. For example, the μCT imaging of the nuclei of adipocytes, fibroblasts, and keratinocytes, or whole pancreatic or hepatocytic cells, could result in data analogous to the hASC nuclei presented in our paper. In addition to the hASC, μCT imaging of the CaCo-2 epithelial cells was demonstrated in Figure 1. Studying the overall size of the cell could be interesting, as the size of the cells can correlate with the phases of the differentiation process, the size can vary in response to physiological stimuli, and also imply about the metabolic state of the cell¹¹. All these aspects are now discussed in the manuscript to motivate a wider scientific audience to consider the possibilities μCT technology can offer.

Comment 2

It would also have been useful to indicate the dimensions of specific biological structures whose sizes determine the resolution needed to see them. A voxel size of 1.2 μm , for example, would

theoretically allow detection of objects of 2.4 μm in size. Therefore, the claim of 1.2 μm voxel resolution needs to be supported by showing resolution 2.4 μm cell or man-made features.

Response 2

We would like to thank the reviewer for this important aspect as well. To detect the presence of a detail, such as an isolated cell structure, the voxel can be much larger than the detail it is representing. To detect the presence of a detail, fundamentally, the only requirement is to have a signal strength that enlightens the representative voxel from the background voxels. However, to resolve two proximate details from each other requires not only enough strong signal but also enough small voxel size and otherwise accurate image data (not heavily affected by image artifacts). To consider the μCT resolution capability achieved in this research, we need to look at the two limit-reaching cases. First, we did not manage to reproduce the curly ZO-1 features (Fig. 1) in the μCT data having a voxel size of 1.2 μm , although the Nyquist sampling theorem suggests the largest curls could be resolved ideally. However, as the subsequent results of the manuscript suggest, resolving objects much less than 2.4 μm apart is possible *in silico*. For example, it was possible to distinguish the expected nuclei from the under-segmented aggregate shown in Figure 3f. Taking into account the principle of how the 3D marker-based watershed separation works, it can be used to resolve objects even in full contact, assuming the markers are correctly placed one per individual object. Thus, although details might not be directly resolvable due to image artifacts, *in silico* processing can improve the resolution of certain structures. This aspect is now better explained in Discussion to help the readers to better understand the possibilities μCT imaging can offer.

Comment 3

The writing can also be more clear. The important concluding sentence of the abstract was particularly puzzling:

“undiscovered potential of the readily available x-ray tomography devices that can be used to study relatively large cell-cultured applications thoroughly with 3D microscopic accuracy.”

What is "relatively large cell-cultured applications thoroughly"?

Response 3

During the final proofreading, we paid further attention to improve the overall clarity of the text. For example, relating to the cited portion of the text, we added the dimensions of the scaffolds in the Abstract.

Comment 4

In sum, the general importance of the work does not seem to be of sufficiently large impact scientifically or methodologically to warrant publication in *Communications Biology*.

Response 4

We respect the opinion. We hope that by the aid of the valuable, constructive comments and criticism the reviewers gave us, we managed to make the paper more interesting for a broader scientific audience. We also hope that as time passes, scientific activity relating to the scope of this

paper increases even more, along with the further development and increasing interest on cell-cultured applications.

Reviewer #2

The authors have undertaken a great effort to improve their ms. They have addressed the points raised in my previous report, and as far as I can see, also those by the other two reviewers in a satisfactory way. The ms meets the high standards of the journal now and should be published. I recommend the present ms for publication in Nature Communications Biology.

Comment 1

side remark: In the first sentence of the Summary section I read "with visually and quantitatively" which should read "visually and quantitatively". Also, this statement occurs more or less twice in the summary section, please change this.

Response 1

We thank the reviewer for the encouraging words and the time and effort given to help us to improve the manuscript. The requested corrections were done.

Reviewer #3

The authors have addressed my concerns. In my opinion, the manuscript is now suitable for publication in Communications Biology.

Response

We thank the author for the valuable help given to help us to improve the manuscript.